# EXPERT DIVERGENCE LEARNING FOR MoE-BASED LANGUAGE MODELS

**Jiaang Li, Haibin Chen, Langming Liu, Yujin Yuan, Yadao Wang, Yizhen Zhang, Chengting Yu, Xin Tong, Weidong Zhang, Shilei Liu, Wenbo Su, Bo Zheng**

Alibaba Group

## ABSTRACT

The Mixture-of-Experts (MoE) architecture is a powerful technique for scaling language models, yet it often suffers from expert homogenization, where experts learn redundant functionalities, thereby limiting MoE's full potential. To address this, we introduce Expert Divergence Learning, a novel pre-training strategy that explicitly encourages functional specialization among experts. Our method incorporates a label-driven auxiliary loss that leverages domain labels inherent in pre-training corpora to maximize the pairwise Jensen-Shannon Divergence between the expert routing distributions of different data domains. This optimization objective allocates the total routing diversity to develop diverged routing policies for varied domains, which leads to well-coordinated expert specialization. We validate our approach by pre-training MoE models of up to 15 billion parameters from scratch. Experimental results demonstrate that models trained with Expert Divergence Learning not only achieve a lower language modeling loss but also exhibit significant performance improvements across a diverse range of downstream benchmarks. Further analysis confirms that our method effectively mitigates expert homogenization and brings greater functional specialization, all with negligible computational overhead during training.

## 1 INTRODUCTION

The Mixture-of-Experts (MoE) architecture (Shazeer et al., 2017; Fedus et al., 2022) has become the de facto standard for scaling Large Language Models (LLMs) (Wei et al., 2022; Achiam et al., 2023; Clark et al., 2022). By sparsely activating a fraction of its feed-forward network (FFN) parameters for each input, an MoE model can match the performance of a much larger dense counterpart at a lower computational cost (Artetxe et al., 2022; Jiang et al., 2024). This paradigm of high capability with reduced FLOPs has cemented MoE's role in many recent state-of-the-art models (Comanici et al., 2025; Team, 2025b; DeepSeek-AI, 2024; OpenAI, 2025).

However, the full potential of MoE is hindered by its standard training paradigm. MoE is naturally designed to *divide and conquer*, allowing experts to develop specialized representations and form diverse combinations to fit heterogeneous real-world data distributions (Kudugunta et al., 2021; Wang et al., 2024a; Qiu et al., 2025). Yet, the current training approach provides no explicit objective to guide this crucial specialization. The sole constraint is typically a load-balancing loss (Lepikhin et al.), which promotes uniform expert usage to ensure routing diversity but lacks a mechanism to guide what each expert should learn. Consequently, experts are often trained on largely overlapping data distributions (Figure 1 (b)), leading to *expert homogenization*—a phenomenon where they develop redundant functionalities and specialize insufficiently (DeepSeek-AI, 2024; Baidu-ERNIE-Team, 2025). This redundancy causes the intended ensemble of specialists to collapse into a group of similar generalists, which is fundamentally at odds with the diverse, multi-domain nature of real-world data that demands expert diverging. This insufficient specialization ultimately diminishes the MoE's effective capacity (Chi et al., 2022; Krishnamurthy et al., 2023; Qiu et al., 2025).

To address this limitation, we argue that effective specialization should not be an emergent property left to chance, but rather a goal that is explicitly guided by an external signal. Building on

---

Correspondence to: Jiaang Li <forgoldenbite@gmail.com>.

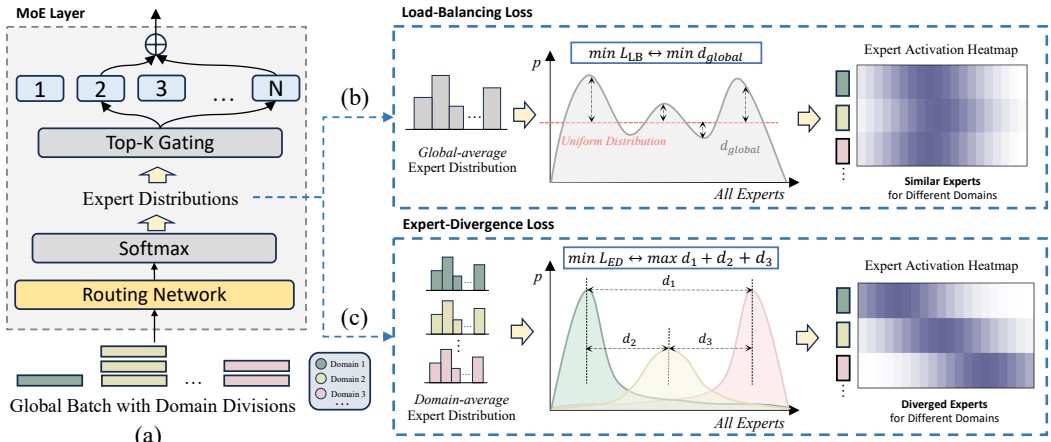

Figure 1: A conceptual comparison of the standard Load-Balancing Loss and our proposed Expert-Divergence Loss. (a) For a batch of tokens partitioned by domain, an MoE layer generates expert routing distributions. (b) The standard Load-Balancing Loss ($\mathcal{L}_{LB}$) promotes uniformity by operating on the global average of expert distributions. This can lead to homogenization, as experts are trained on indistinct data mixtures. (c) In contrast, our Expert-Divergence Loss ($\mathcal{L}_{ED}$) provides a targeted signal by maximizing the divergence between domain-specific average distributions. This guides distinct experts to train on differentiated data subsets and develop functional specialization.

this insight, we introduce **Expert Divergence Learning**, a novel training strategy that incorporates a label-driven auxiliary loss to enforce divergent routing policies across domains. Our method leverages the domain labels, such as source or topic, inherently available in large-scale pre-training corpora to maximize the pairwise Jensen-Shannon Divergence (Lin, 1991) between the average expert routing distributions of different data domains. By maximizing this divergence, our auxiliary loss provides a targeted gradient that complements the load-balancing constraint. While load balancing encourages overall routing diversity, our objective channels this diversity to manifest as clear distinctions between domains, thereby promoting functional specialization and cultivating a more effective and well-coordinated ensemble of specialists.

To validate our approach, we pre-train MoE language models of up to 15 billion parameters from scratch. Results demonstrate that Expert Divergence Learning yields substantial benefits, with a lower language modeling loss and stronger downstream performance across multiple domains. Further analysis indicates enhanced expert specialization, revealing that experts develop distinct functionalities. This work proves the importance of explicitly guiding expert roles during pre-training to unlock more capable MoE-based LLMs.

## 2 RELATED WORKS

**MoE-based Language Models.** Sparsely-gated MoE (Shazeer et al., 2017) has emerged as a key strategy for efficiently scaling Transformer models, demonstrating the power of conditional computation (Lepikhin et al.; Fedus et al., 2022). This paradigm now underpins many state-of-the-art LLMs OpenAI (2025); DeepSeek-AI (2024); Team (2025b;a); Comanici et al. (2025). A central component of MoE training is the load-balancing mechanism. The standard approach involves an auxiliary loss (Cai et al., 2025). Alternative strategies include expert-choice routing (Zhou et al., 2022), where each expert selects tokens based on its capacity, and the use of a router bias term to achieve balance without an explicit loss Wang et al. (2024a). These methods focus on ensuring uniform expert utilization, lacking explicit guidance of what each expert learns.

**Expert Specialization.** While MoE models naturally tend towards expert specialization, a lack of explicit guidance can lead to redundant and overlapping functionalities (Chaudhari et al., 2025; Chen et al., 2022). Some attribute this phenomenon to representational collapse, where tokens cluster around expert centroids (Chaudhari et al., 2025; Chen et al., 2022). To counteract this expert homogenization issue, prior work has focused on architectural modifications. A prevailing strategy involves fixed FNN into MoE, referred to as shared experts, with the goal of capturing commonali-

ties and alleviating parameter redundancy among other routed experts (DeepSeek-AI, 2024; Team, 2024). Besides, Zhong et al. calculate expert merge scores for each sequence and merging all experts into a single expert before computing the corresponding sequence. Although this shows better improvements and specialization, its complex mechanism has not been verified by large-scale training. These structural changes often retain the standard MoE training objectives, and fostering organized collaboration and specialization among these experts is an ongoing challenge.

More recently, attention has shifted towards modifying the training objective. Qiu et al. (2025) has shown that computing the load-balancing loss in the global batch rather than micro-batch can enhance specialization. A concurrent study, ERNIE 4.5 (Baidu-ERNIE-Team, 2025), proposes an unsupervised objective that encourages orthogonality in the router's weight matrix, without considering the domain prior of data structure. Our work fills this gap by introducing a supervised, label-driven loss that explicitly uses domain information to guide specialization.

## 3 METHODOLOGY

In this section, we first review the standard Mixture-of-Experts (MoE) architecture and its training objectives. We then introduce our proposed strategy, **Expert Divergence Learning**, and detail the auxiliary loss designed to foster expert specialization.

### 3.1 BACKGROUND: MOE STRUCTURE AND STANDARD TRAINING OBJECTIVES

A standard MoE layer replaces the feed-forward network (FFN) in a Transformer block with a set of $N$ parallel expert networks $\{E_1, \ldots, E_N\}$ and a routing network. As shown in Figure 1 (a), for each input token $\mathbf{x}$, the router computes logits $\mathbf{h}(\mathbf{x}) = \mathbf{W}_r \mathbf{x}$ and applies a softmax to produce a probability distribution $\mathbf{p}(\mathbf{x})$ over the experts. A Top-K gating mechanism then selects the set of experts $\mathcal{T} = \text{TopK}(\mathbf{p}(\mathbf{x}), \text{K})$ with the highest probabilities. The final output $\mathbf{y}$ is a weighted sum of the selected experts' outputs:

$$\mathbf{y} = \sum_{i \in \mathcal{T}} g_i(\mathbf{x}) E_i(\mathbf{x}), \quad \text{where} \quad g_i(\mathbf{x}) = \frac{p_i(\mathbf{x})}{\sum_{j \in \mathcal{T}} p_j(\mathbf{x})}. \tag{1}$$

The standard pre-training objective, $\mathcal{L}_{\text{standard}}$, combines the primary language modeling loss, $\mathcal{L}_{LM}$, with an auxiliary load-balancing loss, $\mathcal{L}_{LB}$. The language modeling loss is the average negative log-likelihood for a sequence of tokens $s = (x_1, x_2, \ldots, x_T)$:

$$\mathcal{L}_{LM} = -\frac{1}{T} \sum_{t=1}^{T} \log P_\theta(x_t | x_{<t}), \tag{2}$$

where $\theta$ represents the model parameters and $P_\theta(x_t | x_{<t}; \theta)$ is the predicted probability of token $x_t$ given the preceding context $x_{<t}$.

The load-balancing loss is designed to encourage uniform expert usage, as shown in Figure 1 (b). It is implemented as:

$$\mathcal{L}_{LB} = N \cdot \sum_{i=1}^{N} f_i \cdot P_i. \tag{3}$$

For a batch $\mathcal{B}$ containing $T$ tokens, $f_i = \frac{1}{T} \sum_{\mathbf{x} \in \mathcal{B}} \mathbf{1}_{i \in \mathcal{T}_{\mathbf{x}}}$ is the fraction of tokens routed to expert $i$, where $\mathcal{T}_{\mathbf{x}}$ is the set of selected experts for token $\mathbf{x}$, and $P_i = \frac{1}{T} \sum_{\mathbf{x} \in \mathcal{B}} p_i(\mathbf{x})$ is the average router probability for expert $i$. While load-balancing loss, $\mathcal{L}_{LB}$, encourages a global diversity of expert selection across all tokens in a batch, this objective is indiscriminate and does not guide *how* this diversity should be structured.

The final standard objective is a weighted sum:

$$\mathcal{L}_{\text{standard}} = \mathcal{L}_{LM} + \alpha \mathcal{L}_{LB}. \tag{4}$$

## 3.2 Expert Divergence Learning

The standard training objective is agnostic to the semantic or topical domain of the input data. To address this, we introduce a more targeted, fine-grained constraint: we explicitly encourage greater routing diversity between different domains. To explicitly guide experts toward specializing in distinct domains, we introduce Expert Divergence Learning. This strategy incorporates a novel training loss, the Expert Divergence Loss ($\mathcal{L}_{ED}$), which leverages the domain labels readily available in large-scale pre-training corpora. This process is demonstrated in Figure 1 (c).

Let a training batch $\mathcal{B}$ consist of sequences, where each sequence $s = (x_1, x_2, ..., x_T)$ is associated with a domain label $d$ from a set of unique domains $\mathcal{D} = \{1, 2, ..., M\}$. Our method computes $\mathcal{L}_{ED}$ in a three-step process for each MoE layer:

1. **Token-to-Sequence Aggregation**: For each token $x_t$ in a sequence $s$, the router outputs a full probability distribution $p(x_t)$ over the N experts. We first determine the average expert distribution for the entire sequence, denoted as $\overline{p}_s$, by averaging the distributions of all its constituent tokens:

$$\overline{p}_s = \frac{1}{T} \sum_{t=1}^{T} p(x_t). \tag{5}$$

2. **Sequence-to-Domain Aggregation**: Next, we group these sequence-level distributions by their domain labels. For each domain $j \in \mathcal{D}$ present in the batch, we define its subset of sequences as $\mathcal{B}_j = \{s | (s, d) \in \mathcal{B}, d = j\}$. We then compute the average expert probability distribution for domain j, $\overline{p}_j$, by averaging over all sequences within that group:

$$\overline{p}_j = \frac{1}{|\mathcal{B}_j|} \sum_{s \in \mathcal{B}_j} \overline{p}_s. \tag{6}$$

3. **Pairwise Divergence Computation**: The Expert Divergence Loss is designed to maximize the dissimilarity between these average domain-level distributions. We employ the Jensen-Shannon (JS) Divergence for this purpose due to its symmetry and bounded nature. For all unique pairs of domains {j, k} present in the batch, we compute their JS Divergence:

$$D_{JS}(\overline{p}_j || \overline{p}_k) = \frac{1}{2} D_{KL}(\overline{p}_j || m) + \frac{1}{2} D_{KL}(\overline{p}_k || m), \tag{7}$$

where $m = \frac{1}{2}(\overline{p}_j + \overline{p}_k)$ and $D_{KL}(\cdot || \cdot)$ is the Kullback-Leibler (KL) Divergence.

Our final $\mathcal{L}_{ED}$ is defined as the average negative logarithm of the pairwise JS Divergence over all unique pairs of domains present in the batch, which encourages the routing patterns of different domains to diverge from each other:

$$\mathcal{L}_{ED} = \frac{1}{\binom{M_B}{2}} \sum_{\{j,k\} \subseteq \mathcal{D}_B, j<k} -log(D_{JS}(\overline{p}_j || \overline{p}_k) + \epsilon), \tag{8}$$

where $\mathcal{D}_B$ is the set of domains represented in batch B, $M_B = |\mathcal{D}_B|$, and $\epsilon$ is a small constant (e.g., $10^{-8}$) for numerical stability. The negative logarithm in the loss function amplifies the gradient signal when divergence values are small, preventing the vanishing gradient problem and leading to more robust optimization.

This auxiliary loss is integrated into the final training objective with a new hyperparameter $\beta$:

$$\mathcal{L}_{\text{final}} = \mathcal{L}_{LM} + \alpha \mathcal{L}_{LB} + \beta \mathcal{L}_{ED}. \tag{9}$$

This combined objective encourages the model to actively learn distinct routing patterns for data from different domains. As we will demonstrate theoretically in Section 3.3, this approach channels the model's overall routing diversity, which is induced by $\mathcal{L}_{LM}$ and $\mathcal{L}_{LB}$, towards creating divergence between domains rather than within them. This reallocation of diversity gives rise to expert specialization.

### 3.3 THEORETICAL MOTIVATION: FINER-GRAINED DIVERSITY ALLOCATION

The theoretical basis of our method lies in its ability to allocate routing diversity. While $\mathcal{L}_{LB}$ indiscriminately promotes the global diversity in expert selection across all tokens, $\mathcal{L}_{ED}$ provides a finer-grained allocation by explicitly encouraging each domain using diverged experts, as shown in Figure 1. We formalize this concept by decomposing the total routing diversity into two parts.

First, we define the **Total Divergence** ($D_{\text{total}}$) as a measure of the global routing diversity across all tokens. It is calculated as the average KL Divergence between each token's routing distributions $p(x_t)$ and the average distribution for the entire batch, $\overline{p}_{\text{global}}$:

$$D_{\text{total}} = \sum_{\text{all tokens } t} \frac{1}{T} D_{KL}(p(x_t)||\overline{p}_{\text{global}}) = H(\overline{p}_{\text{global}}) - \frac{1}{T} \sum_{\text{all tokens } t} H(p(x_t)). \qquad (10)$$

This total divergence can be decomposed into two parts. The first is the **Inter-Domain Divergence** ($D_{\text{inter}}$), which quantifies the routing diversity *between* different domains. It measures how much the average routing policy for each domain, $\overline{p}_j$, deviates from the global average:

$$D_{\text{inter}} = \sum_{j=1}^{M_B} \frac{T_j}{T} D_{KL}(\overline{p}_j||\overline{p}_{\text{global}}) = H(\overline{p}_{\text{global}}) - \sum_{j=1}^{M_B} \frac{T_j}{T} H(\overline{p}_j). \qquad (11)$$

The second component is the **Intra-Domain Divergence** ($D_{\text{intra}}$), which represents the routing diversity *within* each single domain. It measures the average divergence between a token's routing path and the average distribution of its domain:

$$D_{\text{intra}} = \sum_{j=1}^{M_B} \frac{T_j}{T} \left( \sum_{x_t \in \text{domain } j} \frac{1}{T_j} D_{KL}(p(x_t)||\overline{p}_j) \right) = \sum_{j=1}^{M_B} \frac{T_j}{T} H(\overline{p}_j) - \frac{1}{T} \sum_{\text{all tokens } t} H(p(x_t)). \qquad (12)$$

Full derivations are in Appendix A. Eqs. 10 to 12 directly lead to the following key proposition.

**Proposition 1** (Divergence Decomposition). *The total routing divergence in a batch is the sum of its inter-domain and intra-domain components: $D_{total} = D_{inter} + D_{intra}$.*

This decomposition reveals the distinct roles of the loss functions. The standard load-balancing loss $\mathcal{L}_{LB}$ remains agnostic to how the overall diversity $D_{\text{total}}$ is allocated between its inter- and intra-domain components. In contrast, our proposed expert divergence loss $\mathcal{L}_{ED}$ is explicitly designed to mitigate this issue. As shown in Appendix B, $\mathcal{L}_{ED}$ directly targets and increases the inter-domain divergence $D_{\text{inter}}$ by maximizing pairwise divergence between domains, which leads to our second proposition.

**Proposition 2** (Synergistic Optimization). *Optimizing the Expert Divergence Loss, $\mathcal{L}_{ED}$, promotes an increase in the theoretical inter-domain divergence, $D_{inter}$.*

Therefore, the two losses operate in synergy. $\mathcal{L}_{ED}$ acts as a finer-grained guiding signal that directs the flow of global routing diversity, which is promoted by the load-balancing loss $\mathcal{L}_{LB}$. Specifically, $\mathcal{L}_{ED}$ channels a larger fraction of the total diversity ($D_{\text{total}}$) to be allocated between domains to create meaningful distinctions ($D_{\text{inter}}$), thereby leading to the emergence of specialized experts on each domain.

## 4 EXPERIMENTS

### 4.1 EXPERIMENTAL SETUP

**Model and Implementation Details.** We adopt the Qwen3-MoE (Team, 2025b) model architecture and the tokenizer. Due to computational constraints, we train scaled-down versions of the Qwen3-MoE 30B-A3B model, which has 30 billion non-embedding parameters and activates 3 billion per input. To thoroughly validate our approach on models with different scales, we train models with three varying sizes: 15B-A1.5B, 8B-A0.8B, and 3B-A0.3B. The detailed architectural parameters for each are presented in Table 1. All experiments use 64 GPUs with 80 GB of memory each.

Table 1: Architectural parameters of selected MoE models.

|  | #Layers | hidden-size | intermediate-size | #Experts | Top-K |
|---|---|---|---|---|---|
| **15B-A1.5B** | 16 | 2048 | 1560 | 96 | 8 |
| **8B-A0.8B** | 16 | 2048 | 1344 | 60 | 4 |
| **3B-A0.3B** | 16 | 1536 | 640 | 64 | 4 |

All models are trained using the AdamW optimizer (Loshchilov & Hutter, 2017) with $\beta_1$=0.9, $\beta_2$=0.95 and a weight decay of 0.1. We employ a constant learning rate of $5 \times 10^{-4}$ with a 100-step linear warm-up. For data processing, we use a global batch size of 1 million tokens and pack all inputs to a maximum sequence length of $8192$. For the loss components, the weight for the load-balancing loss $\alpha$ is fixed at $1 \times 10^{-3}$. The expert-divergence loss weight $\beta$ is set to $5 \times 10^{-4}$, a value selected empirically based on training loss stability and performance.

**Training Data.** We pre-train our models from scratch on 100B text tokens. To ensure reproducibility, the training corpus is entirely composed of open-source data. The corpus is a mixture from three sources: 40% from the English content *Nemotron-cc* (Su et al., 2024), 40% from the Chinese content *Fineweb-edu-chinese-v2* (Yu et al., 2025), and 20% from the mathematics content *FineMath* (Allal et al., 2025).

To guide expert specialization, we unpack monolithic pre-training corpora by employing two distinct domain labeling schemes:

- **3-Class Diverging:** This scheme consists of 3 domains, where each sequence is directly labeled with its data source (i.e., English, Chinese, or Math).

- **49-Class Diverging:** To create a more granular set of domains, we categorized our data into 49 distinct topics. We used separate classifiers to label the English and Chinese corpora with 24 topic domains each, following the methodology proposed by Wettig et al. (2025). Mathematics was treated as a single, separate domain, resulting in a total of $24(\text{EN}) + 24(\text{ZH}) + 1(\text{Math}) = 49$ domain labels. A full description of each topic is available in Appendix E, and details on the domain classifiers are provided in Appendix D.

**Evaluation.** We use OpenCompass to evaluate our base models on a wide range of capabilities, including English language capability, Chinese language capability, and mathematical reasoning capability. The evaluation is conducted using 7 benchmarks: *Ceval* (Huang et al., 2023), *MMLU* (Hendrycks et al., 2021), *CMMLU* (Li et al., 2024), *ARC-easy, ARC-challenging* (Clark et al., 2018), *Race-middle*, and *Race-high* (Lai et al., 2017). The former three are evaluated with five-shot while the others are evaluated zero-shot.

## 4.2 MAIN RESULTS

The experimental results, presented in Table 2, confirm the effectiveness of our proposed Expert Divergence Learning strategy. Across all model scales, integrating the label-driven auxiliary loss, $\mathcal{L}_{ED}$, consistently improves performance over the standard MoE training paradigm, validating our hypothesis that explicitly guiding expert specialization is beneficial. Even the coarse-grained, 3-class divergence scheme was sufficient to achieve benchmark gains over the baseline on the larger 8B and 15B model scales.

A key observation is that the performance gains from Expert Divergence Learning scale positively with model size. This trend is most pronounced in our largest model, the 15B-A1.5B, which achieves a final average score of 36.65 under the 49-class scheme—a significant improvement over the baseline's 35.59. While smaller models also show improved average scores, the greater capacity of the 15B model allows it to more effectively translate the benefits of guided specialization into robust performance gains across the evaluation suite. This suggests that as models become larger, they exhibit an emergent ability to convert the structured specialization into substantial performance gains.

Furthermore, analysis of the training dynamics reveals that our approach improves the primary training objective regardless of model size. As illustrated for the 3B-A0.3B model in Figure 2, all

---

https://github.com/open-compass/OpenCompass/

Table 2: Downstream task performance of baseline MoE models versus models trained with Expert Divergence Learning. The best result for each model size is **bolded**.

| Model Configurations | | CEVAL | MMLU | CMMLU | ARC$_e$ | ARC$_c$ | RACE$_m$ | RACE$_h$ | *Avg.* |
|---|---|---|---|---|---|---|---|---|---|
| 15B-A1.5B | MoE | 32.80 | 33.24 | 34.64 | 56.61 | 29.83 | 33.36 | 28.64 | 35.59 |
| | + 3-class *Div.* | 32.53 | 32.98 | 35.16 | **59.08** | **33.56** | 32.94 | 28.10 | 36.34 |
| | + 49-class *Div.* | **33.45** | **33.21** | **35.17** | 57.85 | **33.56** | **34.54** | **28.76** | **36.65** |
| 8B-A0.8B | MoE | 32.88 | **32.54** | 34.22 | 53.62 | 31.53 | 32.52 | 28.42 | 35.10 |
| | + 3-class *Div.* | 33.27 | 31.69 | 33.40 | 56.79 | **32.20** | **32.80** | 28.62 | 35.54 |
| | + 49-class *Div.* | **33.81** | 32.05 | **34.50** | **58.55** | 30.85 | 31.75 | **28.87** | **35.77** |
| 3B-A0.3B | MoE | 32.87 | 31.16 | 32.73 | 50.79 | 31.19 | 31.82 | 28.36 | 34.13 |
| | + 3-class *Div.* | 32.08 | **31.47** | **33.07** | 51.15 | 30.85 | 30.78 | **28.44** | 33.98 |
| | + 49-class *Div.* | **33.75** | 31.41 | 32.91 | **54.50** | **31.53** | 32.03 | 28.27 | **34.91** |

configurations trained with Expert Divergence Learning consistently converge to a lower language modeling loss $\mathcal{L}_{LM}$ than the standard MoE baseline. Altering $\beta$ leads to loss variance, but still robustly better than the standard MoE training. This suggests that our auxiliary loss, $\mathcal{L}_{ED}$, successfully steers the model toward a better optimization landscape for the primary task. The scaling trend observed in the downstream benchmarks suggests that greater model capacity is crucial for converting this fundamental training advantage into superior performance.

Finally, our results suggest a link between the granularity of domain supervision and model performance. Across all three model sizes, the fine-grained 49-class scheme, based on semantic topics, consistently outperforms the coarse-grained 3-class scheme, an advantage supported by both superior downstream task performance and lower final training loss. This consistent advantage leads us to hypothesize that providing more specific, semantically meaningful domain signals is a key factor in cultivating a more effective ensemble of specialized experts. This finding, in turn, points to a promising direction for future MoE optimizations: curating web-scale corpora with fine-grained topical labels appears to be a powerful strategy for unlocking the full potential of sparse models.

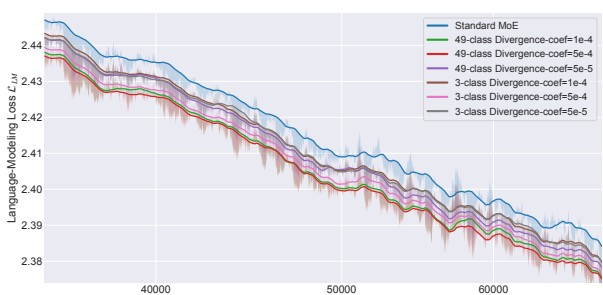

Figure 2: Comparative analysis of the primary language modeling loss $\mathcal{L}_{LM}$ for the 3B-A0.3B models. This figure contrasts the training performance of the baseline MoE with models trained using our Expert Divergence Loss under various configurations, including different domain schemes (3-class and 49-class) and divergence coefficients ($\beta$).

### 4.3 ANALYSIS ON EXPERT SPECIALIZATION

To investigate the mechanism by which Expert Divergence Learning influences expert specialization, we first construct a dedicated validation set. We sample 1,000 instances from each of the three pre-training domains (English, Chinese, and Math), ensuring no overlap with the training data. This resulted in three distinct validation sets: *en_val_1000*, *zh_val_1000*, and *math_val_1000*. Using these sets, we analyze the internal routing behavior of our largest 15B-A1.5B model, comparing the baseline against the version trained with the expert divergence loss.

We conduct a routing perturbation analysis to quantify the functional specialization of experts in each MoE layer by measuring the performance degradation after disrupting the learned routing policy. For each layer $l$ in pre-trained MoE language models, we ablate its learned routing strategy by randomly permuting the row vectors of its router weight matrix, $W_r \in \mathbb{R}^{N \times d}$. Let the matrix be composed of $N$ row vectors, $W_r = [w_1, w_2, \ldots, w_N]^T$, where each row $w_i$ corresponds to the weights for expert $i$. We then create a new matrix $W_r'$ by applying a random permutation $\pi$ to the indices of these row vectors: $W_r' = [w_{\pi(1)}, w_{\pi(2)}, \ldots, w_{\pi(N)}]^T$ This operation shuffles the expert assignments for that layer, forcing tokens to be routed to experts randomly. We then measure the

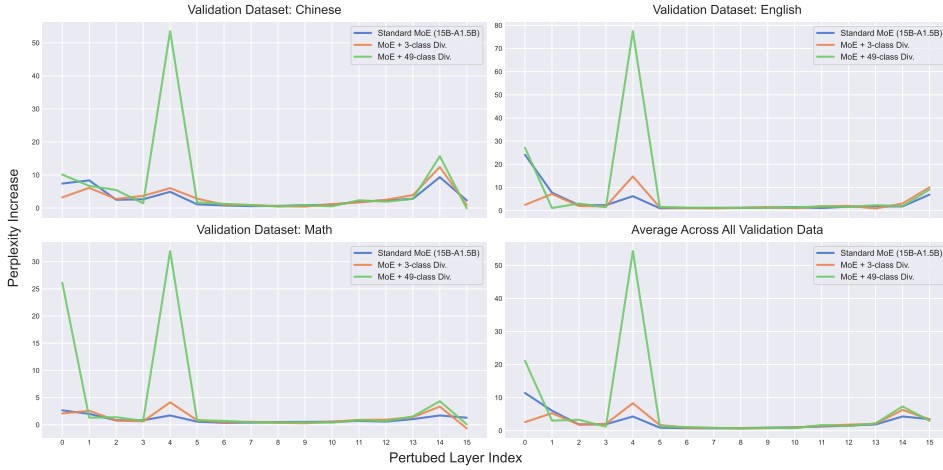

Figure 3: Increase in perplexity ($\Delta PPL$) after randomly permuting router weights for each layer of the pre-trained 15B-A1.5B models. Higher values indicate greater expert specialization.

resulting increase in perplexity ($\Delta PPL^l = PPL^l_{shuffled} - PPL_{original}$) on our validation sets. A small $\Delta PPL^l$ indicates expert homogenization, where experts develop overlapping and redundant functionalities, thereby randomly swapping them would have minimal impact on performance. On the other hand, a large increase signifies that experts have unique, non-interchangeable roles.

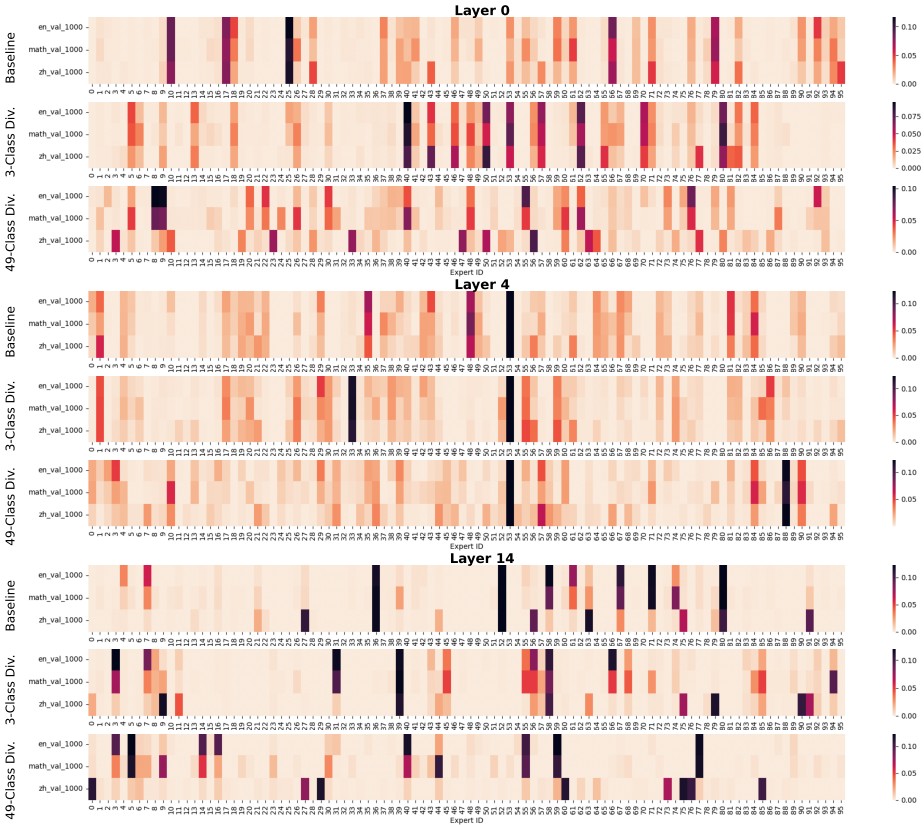

Figure 4: Expert activation heatmaps of different domains for representative layers (0, 4, 14) of the 15B-A1.5B models. Each row shows the average expert activation probabilities for a given validation domain. Darker colors indicate more frequent activation.

The results of $\Delta PPL$ across layers, shown in Figure 3, provide several key insights. First, for all models, random routing in any layer would degrade model performance with different extents,

Table 3: Training throughput (k tokens/s/gpu) across different configurations.

| | 15B-A1.5B | | | 8B-A0.8B | | | 3B-A0.3B | | |
| | MoE | + 3-class *Div.* | + 49-class *Div.* | MoE | + 3-class *Div.* | + 49-class *Div.* | MoE | + 3-class *Div.* | + 49-class *Div.* |
|---|---|---|---|---|---|---|---|---|---|
| TPs | 5.07 | 5.03 | 4.96 | 7.97 | 7.94 | 7.91 | 9.99 | 9.88 | 9.85 |

Table 4: Inference throughput (tokens/s) of 15B-A1.5B models on different domains.

| | en_val_1000 | | | zh_val_1000 | | | math_val_1000 | | |
| | MoE | + 3-class *Div.* | + 49-class *Div.* | MoE | + 3-class *Div.* | + 49-class *Div.* | MoE | + 3-class *Div.* | + 49-class *Div.* |
|---|---|---|---|---|---|---|---|---|---|
| TPs | 14.45 | 14.36 | 14.39 | 19.18 | 19.08 | 19.27 | 19.19 | 19.13 | 19.14 |

indicating the existence of expert specialization. The impact varies significantly across layers, with prominent peaks at specific layers (e.g., 0, 4, and 14). This suggests that expert specialization is concentrated in critical layers rather than being uniformly distributed, consistent with observations that different layers in LLM have different functions (Geva et al., 2020). This natural LLM behavior learned from massive data is not overridden, but rather enhanced by Expert divergence Learning, which steers the optimization process towards a better landscape. Second, models trained with both the 3-class and 49-class divergence mostly exhibit a greater PPL increase than the baseline, particularly at these specialization peaks. This provides direct evidence that our method successfully reduces expert homogenization and makes the expert become more diverse. Moreover, the model trained with the fine-grained 49-class scheme produces a dramatically higher $\Delta PPL$ at Layer 4, far surpassing all other configurations. This result strongly suggests that a more granular supervisory signal induces a higher degree of functional, non-substitutable specialization, compelling the model to develop highly specific roles for its experts.

To more detailed visualize the differences of the routing behaviors, we present expert activation heatmaps for representative layers in Figure 4 (the complete set of heatmaps is available in Appendix F). The average expert activation probabilities for each domain validation set are computed to generate these heatmaps. These resulting vectors are then normalized to sum to 1 for visualization and plotted to reveal the expert utilization patterns. These heatmaps provide a clear qualitative confirmation of our findings from the perturbation experiment. For instance, in layers with a high $\Delta PPL$ such as Layer 4, the model trained with 49-class divergence exhibits markedly more distinct domain-specific activation patterns, revealing a less-overlapping set of preferred experts for each domain. In contrast, the baseline model's activations for different domains are more overlapping.

In summary, this analysis demonstrates that Expert Divergence Learning effectively amplifies the natural specialization tendencies within MoE models. By encouraging divergent routing policies, our method guides the model to develop experts with stronger functional specificity in key layers, which in turn drives the performance improvements reported in Section 4.2.

## 4.4 EFFICIENCY ANALYSIS

We evaluated the computational overhead of Expert Divergence Learning by measuring both training and inference throughput, targeting potential inefficiencies from the $\mathcal{L}_{ED}$ loss calculation and potential latency from expert specialization during domain-specific inference. As shown in Tables 3 and 4, our method introduces negligible overhead in either setting. The training throughput remains comparable to the baseline because the $\mathcal{L}_{ED}$ calculation is lightweight, operating on small-dimensional router outputs. Similarly, inference speed on our largest 15B-A1.5B model is maintained even on domain-specific data, indicating that expert specialization does not create significant expert imbalance that leads to latency. These results confirm that the performance and specialization benefits of our method are achieved without compromising computational efficiency.

## 5 CONCLUSION

In this paper, we introduced Expert Divergence Learning, a novel learning objective that mitigates expert homogenization in MoE models by maximizing routing divergence between domains. Our approach is grounded in the theoretical insight that total routing diversity can be decomposed, allow-

ing our method to channel this diversity specifically toward creating inter-domain distinctions. Experiments demonstrate that this strategy yields a lower language modeling loss and superior downstream performance, particularly for larger models, all with negligible computational cost. Further analysis confirms that these performance gains are driven by enhanced expert specialization, as the guided routing trains experts on more differentiated data mixtures to develop distinct functionalities.

This work validates the importance of explicitly guiding expert roles during the pre-training of sparse models and demonstrates that leveraging the inherent domain structure of web-scale corpora is a powerful and efficient strategy for advancing MoE capabilities. Future directions include scaling LLMs with more sophisticated properties and connections curated from massive data.

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

## A  DERIVATIONS FOR DIVERGENCE DECOMPOSITION

This appendix provides detailed, step-by-step derivations for the three key components of our divergence decomposition identity, $D_{\text{inter}} + D_{\text{intra}} = D_{\text{total}}$. We begin by deriving the entropy-based formula for the generalized Jensen-Shannon Divergence, which is fundamental to all three components.

### A.1  DERIVATION OF THE GENERALIZED JSD FORMULA

The generalized JSD for a set of distributions $\{p_k\}$ with weights $\{\pi_k\}$ is defined as the weighted average of the Kullback-Leibler (KL) divergence from each distribution to their collective mean, $\overline{p} = \sum_k \pi_k p_k$. The derivation from its KL-based definition to the more intuitive entropy-based formula is as follows:

$$
\begin{aligned}
\text{JSD}_\pi(\{p_k\}) &= \sum_k \pi_k D_{KL}(p_k || \overline{p}) && \text{(1. Fundamental Definition)} \\
&= \sum_k \pi_k \left( \sum_i p_{k,i} \log \frac{p_{k,i}}{\overline{p}_i} \right) && \text{(2. Expand KL Divergence)} \\
&= \sum_k \pi_k \left( \sum_i p_{k,i} (\log p_{k,i} - \log \overline{p}_i) \right) && \text{(3. Use log property)} \\
&= \sum_k \pi_k \left( \sum_i p_{k,i} \log p_{k,i} - \sum_i p_{k,i} \log \overline{p}_i \right) && \text{(4. Distribute the term } p_{k,i}) \\
&= \sum_k \pi_k \left( -H(p_k) - \sum_i p_{k,i} \log \overline{p}_i \right) && \text{(5. Use Entropy definition)} \\
&= -\sum_k \pi_k H(p_k) - \sum_k \pi_k \sum_i p_{k,i} \log \overline{p}_i && \text{(6. Distribute the outer weight } \pi_k) \\
&= -\sum_k \pi_k H(p_k) - \sum_i \left( \sum_k \pi_k p_{k,i} \right) \log \overline{p}_i && \text{(7. Swap the order of summation)} \\
&= -\sum_k \pi_k H(p_k) - \sum_i \overline{p}_i \log \overline{p}_i && \text{(8. Substitute def. of } \overline{p}_i) \\
&= -\sum_k \pi_k H(p_k) - (-H(\overline{p})) && \text{(9. Use Entropy definition again)} \\
&= H(\overline{p}) - \sum_k \pi_k H(p_k) && \text{(10. Final Form)}
\end{aligned}
$$

This final expression, $H(\text{mean distribution}) - \text{mean}(H(\text{distributions}))$, forms the basis for the subsequent derivations.

### A.2  APPLICATION TO $D_{\text{TOTAL}}$, $D_{\text{INTER}}$, AND $D_{\text{INTRA}}$

Using the general formula derived above, we can now specify it for the three components in our main argument. We assume the lengths of all inputs are equal. Then the sequence-level average distribution is equal to token-level average distribution, which is directly used in our derivations.

#### A.2.1  TOTAL DIVERGENCE ($D_{\text{TOTAL}}$)

The total divergence is the generalized JSD applied to the set of all $T$ individual token-level distributions $\{p(x_t)\}$ in the batch, each with a uniform weight of $1/T$. Their collective mean is the global

average distribution, $\overline{p}_{\text{global}}$.

$$D_{\text{total}} = \text{JSD}_{1/T}(\{p(x_t)\})$$

$$= H(\overline{p}_{\text{global}}) - \sum_{t=1}^{T} \frac{1}{T} H(p(x_t))$$

$$= H(\overline{p}_{\text{global}}) - \frac{1}{T} \sum_{t=1}^{T} H(p(x_t))$$

### A.2.2 INTER-DOMAIN DIVERGENCE ($D_{\text{INTER}}$)

The inter-domain divergence is the generalized JSD applied to the set of $M_B$ domain-mean distributions $\{\overline{p}_j\}$, where each distribution is weighted by its relative size in the batch, $\pi_j = T_j/T$. The collective mean of these is also $\overline{p}_{\text{global}}$.

$$D_{\text{inter}} = \text{JSD}_{T_j/T}(\{\overline{p}_j\})$$

$$= H(\overline{p}_{\text{global}}) - \sum_{j=1}^{M_B} \frac{T_j}{T} H(\overline{p}_j)$$

### A.2.3 INTRA-DOMAIN DIVERGENCE ($D_{\text{INTRA}}$)

The intra-domain divergence is defined as the weighted average of the JSD *within* each domain. For a single domain $j$, the intra-domain JSD is computed over its $T_j$ token distributions.

$$\text{JSD}(\text{domain } j) = H(\overline{p}_j) - \frac{1}{T_j} \sum_{x_t \in \text{domain } j} H(p(x_t))$$

$D_{\text{intra}}$ is the average of these values, weighted by $\pi_j = T_j/T$.

$$D_{\text{intra}} = \sum_{j=1}^{M_B} \frac{T_j}{T} (\text{JSD}(\text{domain } j)) \qquad \text{(1. Definition)}$$

$$= \sum_{j=1}^{M_B} \frac{T_j}{T} \left( H(\overline{p}_j) - \frac{1}{T_j} \sum_{x_t \in j} H(p(x_t)) \right) \qquad \text{(2. Substitute JSD formula)}$$

$$= \sum_{j=1}^{M_B} \left( \frac{T_j}{T} H(\overline{p}_j) - \frac{1}{T} \sum_{x_t \in j} H(p(x_t)) \right) \qquad \text{(3. Distribute the weight)}$$

$$= \sum_{j=1}^{M_B} \frac{T_j}{T} H(\overline{p}_j) - \frac{1}{T} \sum_{j=1}^{M_B} \sum_{x_t \in j} H(p(x_t)) \qquad \text{(4. Split into two summations)}$$

$$= \sum_{j=1}^{M_B} \frac{T_j}{T} H(\overline{p}_j) - \frac{1}{T} \sum_{\text{all tokens } t} H(p(x_t)) \qquad \text{(5. Combine inner summations)}$$

With these three expanded forms, one can directly verify the decomposition identity: $D_{\text{inter}} + D_{\text{intra}} = D_{\text{total}}$.

## B PROOF OF PROPOSITION 2 (SYNERGISTIC OPTIMIZATION)

This section formally proves that minimizing our proposed Expert Divergence Loss ($\mathcal{L}_{ED}$) synergistically promotes the increase of the theoretical inter-domain divergence ($D_{\text{inter}}$). We first provide an intuitive geometric explanation. Then, we rigorously establish this by using a second-order Taylor expansion. We show that the sum of pairwise divergences ($S_{\text{pair}} = \sum_{j<k} D_{JS}(\overline{p}_j, \overline{p}_k)$) and $D_{\text{inter}}$ are locally proportional ($S_{\text{pair}} \propto D_{\text{inter}}$). A direct proportionality guarantees a monotonic relationship, meaning they increase or decrease together. In practice, we observe that the stronger optimization signal on individual pairs of $\mathcal{L}_{ED}$ brings more training gains while directly using $D_{inter}$ falls behind.

## B.1 INTUITIVE EXPLANATION.

We begin with an intuitive explanation. If we view each domain's expert distribution as a point in a high-dimensional space, the theoretical objective ($D_{\text{inter}}$) measures the spread of these points from their collective center of mass. Our practical objective, which maximizes the sum of all pairwise distances ($S_{\text{pair}}$), imposes a stricter condition: it pushes every point away from every other point. Maximizing all pairwise distances inherently increases the distance from the center, thus providing a strong optimization pressure towards increasing $D_{\text{inter}}$.

## B.2 PROOF STRATEGY.

The goal is to prove that increasing $S_{\text{pair}}$ promotes an increase in $D_{\text{inter}}$. We will establish this by showing that the two quantities are approximately proportional in the local region ($S_{\text{pair}} \propto D_{\text{inter}}$). We will achieve this by showing that:

1. The second-order Taylor expansions of both measures are directly proportional.

2. The perturbation term in this expansion is small enough in our practical setting to render higher-order remainder terms negligible.

This establishes a monotonic relationship, confirming that maximizing the sum of pairwise divergences (the objective of $\mathcal{L}_{ED}$) serves as an effective proxy for maximizing the overall inter-domain divergence.

## B.3 DERIVATION.

For notational simplicity, we assume the weights of all $M_B$ domains within the batch are equal, i.e., $\pi_j = 1/M_B$. Let the shift of each domain distribution from the mean be $\delta_j = \overline{p}_j - \overline{p}_{\text{global}}$, which are centered such that $\sum_j \delta_j = 0$.

### B.3.1  1. APPROXIMATING GLOBAL DISPERSION ($D_{\text{INTER}}$):

The global dispersion is defined as $D_{\text{inter}} = H(\overline{p}_{\text{global}}) - \frac{1}{M_B} \sum_j H(\overline{p}_j)$. To approximate this, we first expand the average entropy term $\frac{1}{M_B} \sum_j H(\overline{p}_j)$:

$$\frac{1}{M_B} \sum_j H(\overline{p}_j) = \frac{1}{M_B} \sum_j H(\overline{p}_{\text{global}} + \delta_j)$$
$$\approx \frac{1}{M_B} \sum_j \left[ H(\overline{p}_{\text{global}}) + \nabla H^T \delta_j + \frac{1}{2} \delta_j^T \nabla^2 H \delta_j \right]$$
$$= H(\overline{p}_{\text{global}}) + \nabla H^T \left( \frac{1}{M_B} \sum_j \delta_j \right) + \frac{1}{2M_B} \sum_j \delta_j^T \nabla^2 H \delta_j$$

Since the mean shift $\sum_j \delta_j = 0$, the linear term vanishes. Substituting this back into the definition of $D_{\text{inter}}$:

$$D_{\text{inter}} \approx H(\overline{p}_{\text{global}}) - \left[ H(\overline{p}_{\text{global}}) + \frac{1}{2M_B} \sum_j \delta_j^T \nabla^2 H \delta_j \right]$$
$$= -\frac{1}{2M_B} \sum_j \left( \delta_j^T \nabla^2 H \delta_j \right)$$

### B.3.2   2. APPROXIMATING PAIRWISE DISPERSION ($S_{\text{PAIR}}$):

The sum of pairwise divergences is $S_{\text{pair}} = \sum_{j<k} D_{JS}(\overline{p}_j, \overline{p}_k)$, where $D_{JS}(\overline{p}_j, \overline{p}_k) = H\left(\frac{\overline{p}_j + \overline{p}_k}{2}\right) - \frac{1}{2}H(\overline{p}_j) - \frac{1}{2}H(\overline{p}_k)$. We expand each of the three entropy terms around $\overline{p}_{\text{global}}$:

$$\text{(a) } H\left(\frac{\overline{p}_j + \overline{p}_k}{2}\right) = H\left(\overline{p}_{\text{global}} + \frac{\delta_j + \delta_k}{2}\right)$$

$$\approx H(\overline{p}_{\text{global}}) + \frac{1}{2}\nabla H^T(\delta_j + \delta_k) + \frac{1}{8}(\delta_j + \delta_k)^T\nabla^2 H(\delta_j + \delta_k)$$

$$\text{(b) } -\frac{1}{2}H(\overline{p}_j) \approx -\frac{1}{2}\left[H(\overline{p}_{\text{global}}) + \nabla H^T\delta_j + \frac{1}{2}\delta_j^T\nabla^2 H\delta_j\right]$$

$$\text{(c) } -\frac{1}{2}H(\overline{p}_k) \approx -\frac{1}{2}\left[H(\overline{p}_{\text{global}}) + \nabla H^T\delta_k + \frac{1}{2}\delta_k^T\nabla^2 H\delta_k\right]$$

Summing these three parts, the constant and linear terms cancel out, leaving only the quadratic (Hessian) terms:

$$D_{JS}(\overline{p}_j, \overline{p}_k) \approx \frac{1}{8}(\delta_j + \delta_k)^T\nabla^2 H(\delta_j + \delta_k) - \frac{1}{4}(\delta_j^T\nabla^2 H\delta_j + \delta_k^T\nabla^2 H\delta_k)$$

$$= -\frac{1}{8}(\delta_j - \delta_k)^T\nabla^2 H(\delta_j - \delta_k)$$

Summing over all pairs, we can get $S_{\text{pair}}$. Since $\sum_j \delta_j = 0$, we have the identity $\sum_{j<k}(\delta_j - \delta_k)^T A(\delta_j - \delta_k) = M_B \sum_j \delta_j^T A\delta_j$ for a symmetric matrix $A$. Applying this identity, we have:

$$S_{\text{pair}} \approx -\frac{1}{8}\sum_{j<k}(\delta_j - \delta_k)^T\nabla^2 H(\delta_j - \delta_k)$$

$$= -\frac{M_B}{8}\sum_j \delta_j^T\nabla^2 H\delta_j$$

### B.3.3   CONCLUSION.

By comparing the two approximations, we find a direct proportionality between their dominant second-order terms:

$$S_{\text{pair}} \approx \left(\frac{M_B^2}{4}\right) D_{\text{inter}}$$

### B.4   JUSTIFICATION OF THE PROPORTIONAL APPROXIMATION

The above derivation shows that the principal components of the Taylor expansions for $S_{pair}$ and $D_{inter}$ are positively proportional. We further show that the higher-order remainder terms of the expansion are relatively negligible and don't affect this proportionality.

In our MoE training context, the perturbation vectors, $\delta_j$, are indeed kept small. This is a consequence of two key factors: the **architectural design**, which distributes probability mass across a large number of experts ($N \geq 60$), and the **load-balancing regularization** ($\mathcal{L}_{LB}$), which explicitly penalizes routing concentration. Together, these ensure that the components of the perturbation vector, $|\delta_{j,i}|$, are significantly less than 1.

Because the perturbations are small, the higher-order terms of the Taylor series (e.g., $O(\delta_j^3)$) diminish at a super-linear rate and have a negligible impact relative to the second-order term. Therefore, we can confidently approximate the relationship between the two functions by the proportionality of their dominant second-order terms. This establishes a local monotonic relationship, confirming that an optimization step increasing $S_{pair}$ will also increase $D_{inter}$. This validates the theoretical foundation of our Expert Divergence Learning method as claimed in Proposition 2.   □

## C   TRAINING PROCEDURE OF EXPERT DIVERGENCE LEARNING

The pseudo code of one training step can be found in Algorithm 1.

---

**Algorithm 1** Training Step with Expert Divergence Learning

---

**Require:** Model parameters $\theta$, hyperparameters $\alpha, \beta$.

1: Sample a batch of sequence-domain pairs $\mathcal{B} = \{(s_k, d_k)\}_{k=1}^{|\mathcal{B}|}$.
2: Unpack all sequences into a flat list of tokens $\{x_i\}_{i=1}^{T}$.
    **1. Forward Pass**
3: Logits, $\{p(x_i)\}_{i=1}^{T} \leftarrow \text{Model}_\theta(\{x_i\})$             ▷ Get LM logits and router probabilities
4: $\mathcal{L}_{\text{LM}} \leftarrow \text{CrossEntropyLoss(Logits)}$
5: $\mathcal{L}_{\text{LB}} \leftarrow \text{ComputeLoadBalancingLoss}(\{p(x_i)\}_{i=1}^{T})$
    **2. Compute Expert Divergence Loss**
6: *//Aggregate token-level probabilities to sequence-level*
7: For each sequence $s_k \in \mathcal{B}$: $\bar{p}_{s_k} \leftarrow \frac{1}{T_k} \sum_{t=1}^{T_k} p(x_t^k)$
8: *//Aggregate sequence-level probabilities to domain-level*
9: Let $\mathcal{D}_\mathcal{B}$ be the set of unique domains in $\mathcal{B}$.
10: For each domain $j \in \mathcal{D}_\mathcal{B}$: $\bar{p}_{d_j} \leftarrow \frac{1}{|\mathcal{B}_j|} \sum_{s_k \in \mathcal{B}_j} \bar{p}_{s_k}$
11: *//Vectorized computation of pairwise divergence*
12: Let $\mathbf{P}_D \in \mathbb{R}^{|\mathcal{D}_\mathcal{B}| \times N}$ be the matrix of stacked domain distributions $[\bar{\mathbf{p}}_{d_1}, \bar{\mathbf{p}}_{d_2}, \dots]^\top$.
13: **JSD_Matrix** $\leftarrow$ -log(VectorizedPairwiseJSD($\mathbf{P}_D$))      ▷ Results in a $|\mathcal{D}_\mathcal{B}| \times |\mathcal{D}_\mathcal{B}|$ matrix
14: $\mathcal{L}_{\text{ED}} \leftarrow \text{Mean(UpperTriangle(}\mathbf{JSD\_Matrix}))$
    **3. Update Parameters**
15: $\mathcal{L}_{\text{final}} \leftarrow \mathcal{L}_{\text{LM}} + \alpha \mathcal{L}_{\text{LB}} + \beta \mathcal{L}_{\text{ED}}$
16: $\nabla_\theta \mathcal{L}_{\text{final}} \leftarrow \text{backward}(\mathcal{L}_{\text{final}})$
17: $\theta \leftarrow \text{OptimizerStep}(\theta, \nabla_\theta \mathcal{L}_{\text{final}})$

---

# D    DOMAIN CLASSIFIER DETAILS

To create a granular set of domains for our experiments, we employed two separate classifiers to categorize the English and Chinese content of our pre-training corpus into distinct topics.

**English Classifier.**    We directly utilize the classifier developed by Wettig et al. (2025) for English content. This model is initialized from `gte-base-en-v1.5` (Li et al., 2023) and fine-tuned in two stages: first with one million annotations generated by Llama-3.1-8B-Instruct, followed by a second stage of fine-tuning on 80,000 high-quality annotations from Llama-3.1-405B-Instruct.

**Chinese Classifier.**    We trained the Chinese classifier in-house. To generate a training dataset, we translated the annotation prompt from Wettig et al. (2025) into Chinese and used it to collect 500,000 annotations with DeepSeek-r1. We then used these annotations to fine-tune the `chinese-roberta-wwm-ext-large` model (Cui et al., 2021), which serves as our final Chinese domain classifier.

# E    DOMAIN DESCRIPTIONS

This section details the topic schema used for our 49-class divergence experiments. To create fine-grained, semantically meaningful domain labels, we adopted the 24-topic classification system proposed by Wettig et al. (2025). The description and scope of each topic are provided in the table 5.

# F    EXPERT ACTIVATION HEATMAPS

In figure 5, we provide the complete expert activation heatmaps for all MoE layers in our 15B-A1.5B models, offering a comprehensive view of their routing behaviors. Across all layers, models trained with Expert Divergence Learning, particularly the 49-class variant, exhibit more distinct and diversified expert utilization patterns for different domains compared to the baseline.

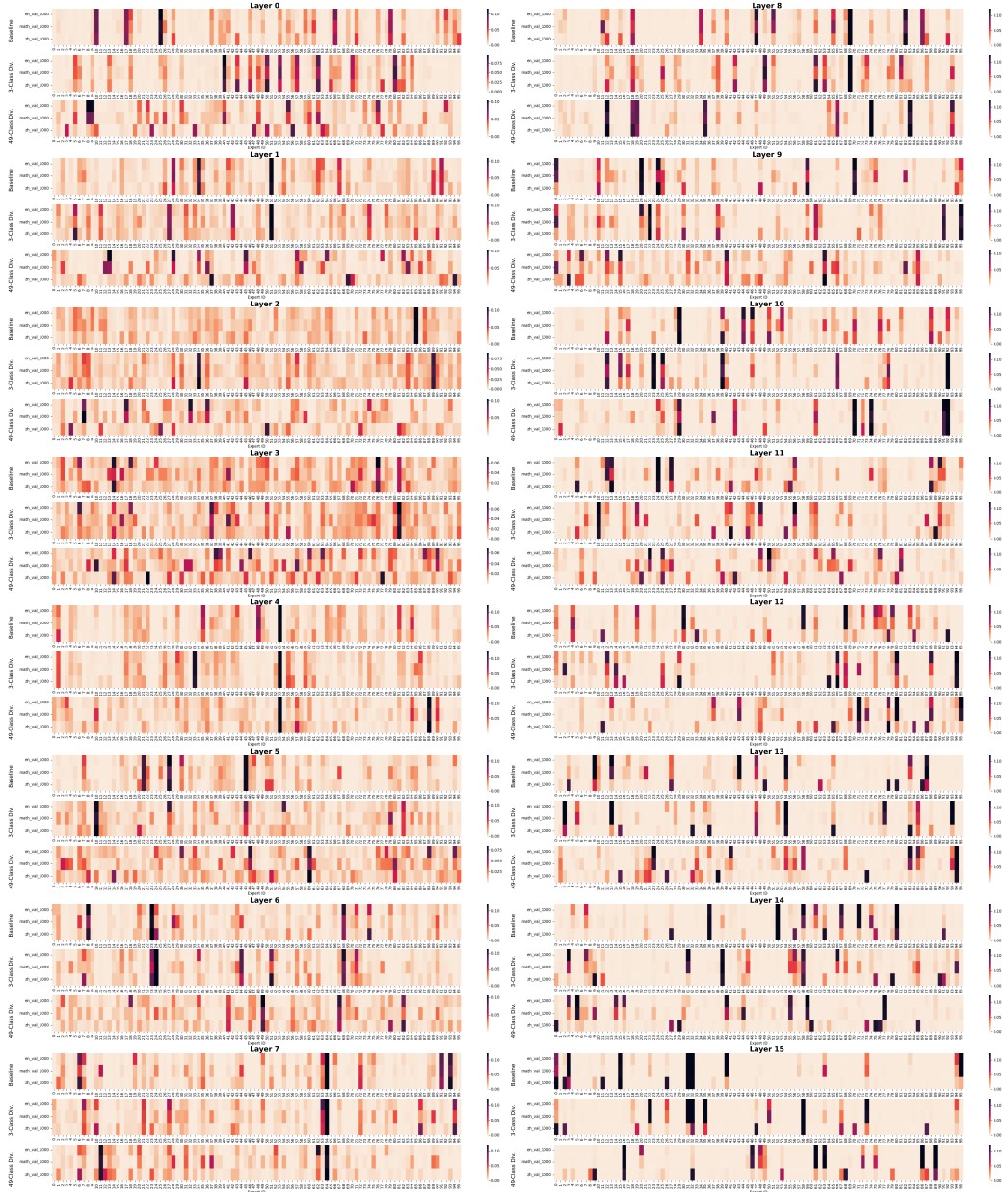

Figure 5: Expert activation heatmap of all layers in 15B-A1.5B models.

Table 5: Domain schema for the 49-class divergence experiments. The 24 topics listed below were applied independently to both the English and Chinese corpora. Mathematics was treated as a distinct, 25th topic, resulting in a total of $24(EN) + 24(ZH) + 1(Math) = 49$ unique domain labels.

| Topic | Description |
|---|---|
| Adult | General adult-oriented content. |
| Art & Design | Includes: architecture. |
| Crime & Law | Includes: law enforcement. |
| Education & Jobs | Includes: pedagogy, training, certification, and academic topics. |
| Entertainment | Includes: music, movies, TV shows, videos, celebrities, humor, and nightlife. |
| Fashion & Beauty | Includes: clothing, accessories, and cosmetics. |
| Finance & Business | Includes: taxes, regulations, investments, insurance, credit, personal finance, and corporate matters. |
| Food & Dining | Includes: recipes, groceries, beverages, and restaurants. |
| Games | Includes: video games, board games, and gambling. |
| Hardware | Includes: computer hardware, phones, televisions, and other consumer electronics. |
| Health | Includes: medicine, wellness, mental health, and veterinary science. |
| History | Includes: geography and archaeology. |
| Home & Hobbies | Includes: real estate, renting, furniture, home improvement, DIY, gardening, pets, toys, and collecting. |
| Industrial | Includes: topics related to manufacturing, utilities, and construction. |
| Literature | Includes: literary criticism, linguistics, philosophy, and related subjects in the humanities. |
| Politics | Includes: social issues, political campaigns, the legislative process, geopolitics, and activism. |
| Religion | Includes: spirituality. |
| Science & Technology | Includes: physics, chemistry, biology, environmental science, mathematics, statistics, and biotech. |
| Social Life | Includes: family, friends, relationships, and community. |
| Software | Includes: topics related to the use of software and the internet. |
| Software Development | Includes: algorithms, coding, and web development. |
| Sports & Fitness | Includes: martial arts, motor sports, outdoor activities, and sports equipment. |
| Transportation | Includes: cars and other vehicles, taxis, public transportation, traffic, commuting, aviation, rail, shipping, logistics. |
| Travel | Includes: hospitality, hotels, cruises and sight-seeing. |

## G USE OF LARGE LANGUAGE MODELS DURING PAPER WRITING

During the preparation of this manuscript, we used a large language model to polish our writing. Its primary role was to refine the text that we had already written, including the descriptions of our methodology and the presentation of mathematical derivations. This was done to improve the clarity and readability of the paper and hopefully make it easier for the readers to understand. We've carefully checked that the LLM-generated texts to make sure there are no contents that are inconsistent with our initial intents.

## H ADDITIONAL TRAINING DYNAMICS AND SPECIALIZATION ANALYSIS

### H.1 TRAINING CURVES TO 100B TOKENS

We present the full training loss trajectories extending to 100B tokens of all model scales in Figure 6. The curves demonstrate that our Expert Divergence Learning method consistently achieves lower language-modeling loss compared to the standard MoE baseline throughout the entire pre-training process.

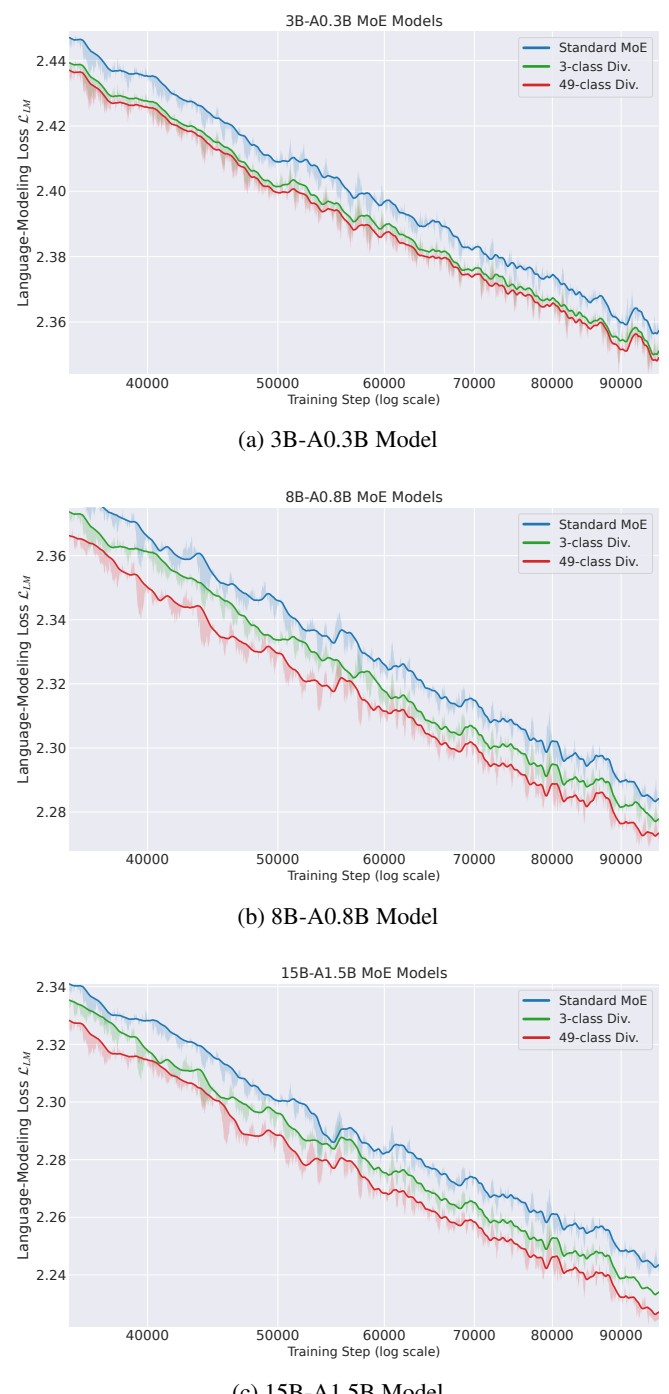

Figure 6: Training Loss Curves to 100B Tokens. Comparison of the language modeling loss ($\mathcal{L}_{LM}$) curves for 3B, 8B, and 15B MoE models.

## H.2 INVERSE EXPERT ACTIVATION HEATMAP AND TERNARY SIMPLEX ANALYSIS

To investigate expert specialization from an expert-centric perspective, we visualize the inverse conditional probability $P(\text{Domain} \mid \text{Expert})$ in Figure 7. These plots illustrate the domain composition of tokens processed by each expert. The results indicate that our method, particularly the 49-class divergence variant, significantly enhances expert specialization. The token distribution for each

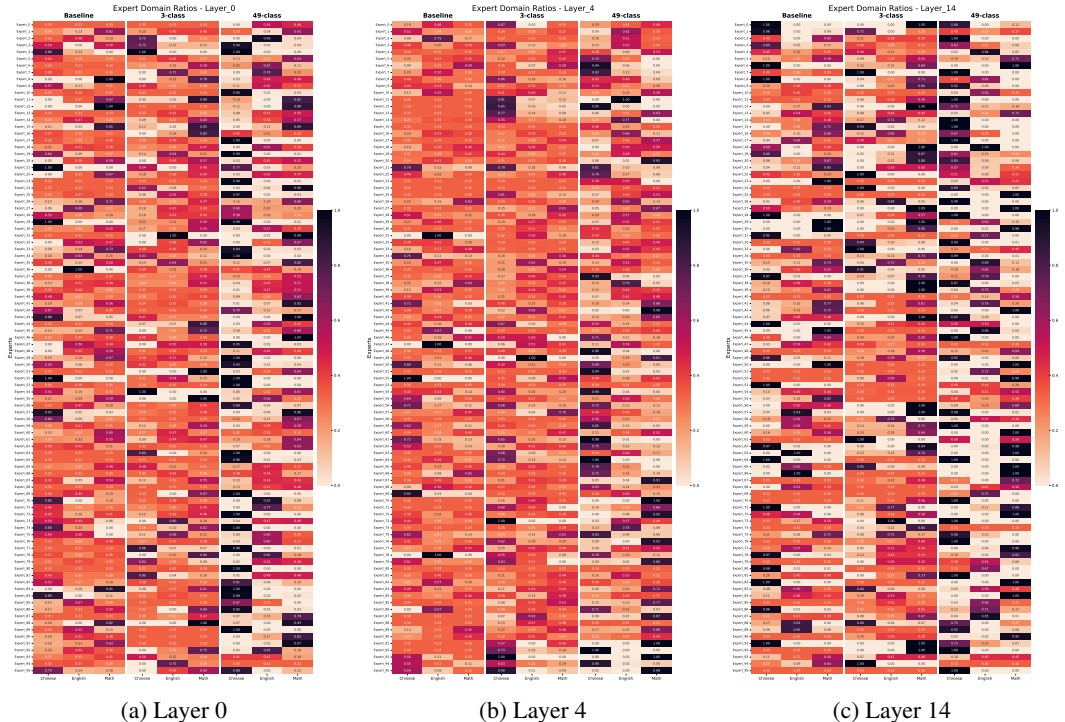

(a) Layer 0          (b) Layer 4          (c) Layer 14

Figure 7: Inverse expert activation heatmaps of different domains for representative layers (0, 4, 14) of the 15B-A1.5B models. Each row shows how often an expert is selected for a particular domain. Darker colors indicate higher frequency.

expert becomes highly concentrated on specific domains rather than being uniformly distributed, confirming that experts have evolved into specialized modules rather than remaining as generalists.

To further provide a holistic view of the global specialization landscape, we project the $P(\text{Domain} \mid \text{Expert})$ distributions onto a ternary simplex, forming a triangle plot in Figure 8. In this visualization, points near the vertices represent "pure" specialists that exclusively process inputs from a single domain, while points near the centroid represent generalists processing all domains equally. This global visualization reveals a striking contrast: while baseline experts cluster near the center (indicating generalist behavior), our method effectively pushes experts toward the distinct domain corners. This trend is consistent across layers, visually confirming that our experts have transformed into functionally specialized units.

# I    STABILITY AND GRANULARITY ANALYSIS

To demonstrate the statistical significance of our improvements and potential fluctuations in evaluation results, we conducted a rigorous stability analysis. We evaluated our largest model (15B-A1.5B) across four distinct late-stage training checkpoints (94B, 96B, 98B, and 100B tokens).

As detailed in Table 6, both the 3-class and 49-class Expert Divergence Learning methods consistently outperform the standard MoE baseline at every single checkpoint. We observe the following trends.

- **Low Variance:** The standard deviation across these checkpoints is low ($\sigma \approx 0.13 - 0.21$), indicating stable convergence.
- **Significant Gap (Ours vs. baseline):** The average performance improvement of our 49-class model over the baseline is $1.10$ points ($36.69 - 35.59$), which significantly exceeds the standard deviation ($1.10 > 3\sigma$). This satisfies the $3\sigma$ criterion for statistical significance, confirming that the observed gains are robust and not due to random training noise.
- **Impact of Finer Granularity:** While the 3-class scheme is competitive, the 49-class setting achieves a higher checkpoint-average score ($36.69$ vs. $36.36$) and wins on the majority of

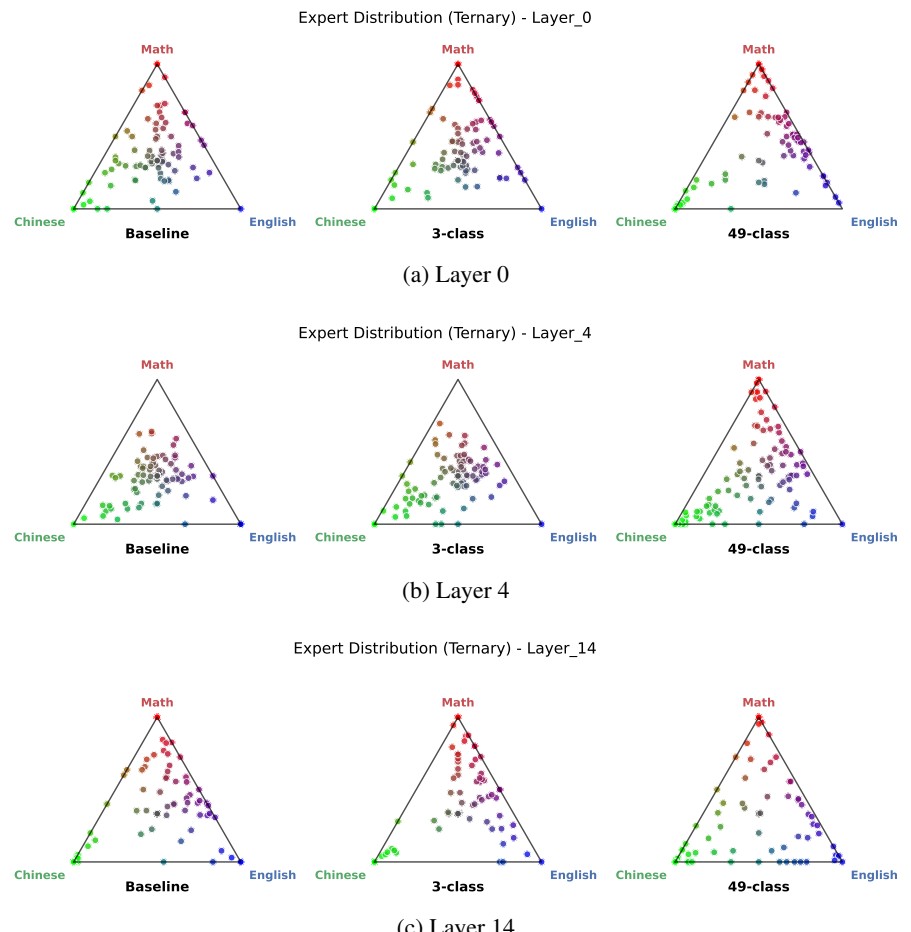

Figure 8: Ternary simplex plots showing the expert distribution across domains (English, Chinese, Math) for representative layers. Points near corners represent specialized experts, while points near the center represent generalists.

benchmarks (6 out of 7). Furthermore, the 49-class model achieves the lowest final training loss across all model scales (Figure 6), suggesting it provides a better optimization landscape. To investigate the upper limit of granularity, we further perform an ablation treating each document as a unique domain. This yields a training loss trajectory slightly worse than the 49-class setting. This indicates a trade-off: extreme granularity maximizes repulsion but penalizes routing consistency between semantically similar documents. Thus, semantic grouping (as in our 49-class scheme) strikes a better balance.

## J    COMPATIBILITY WITH ADVANCES IN MOE

We have validated our proposed Expert Divergence Learning within the standard and widely adopted MoE architecture and training pipeline. It serves as a flexible training objective that is orthogonal to—and can be effectively combined with—other recent advancements in MoE architectures and training strategies.

**1.    Shared-Expert Architectures.**    Recent architectures often employ "shared experts" that are always activated to capture common knowledge, alongside "routed experts" for specialized tasks (DeepSeek-AI, 2024). Our method is highly compatible with this design. While shared experts handle universal features, $\mathcal{L}_{ED}$ can be explicitly applied to the routed experts to maximize their functional distinctiveness. By enforcing divergence among the routed experts, our method fur-

Table 6: **Stability Analysis of 15B-A1.5B Models.** Performance comparison across late-stage training checkpoints (94B to 100B tokens). The results demonstrate consistent improvements with low variance, confirming the statistical significance of our method.

| Model | Checkpoint | C-Eval | MMLU | CMMLU | ARC-e | ARC-c | RACE-m | RACE-h | *Avg.* |
|---|---|---|---|---|---|---|---|---|---|
| **Baseline MoE** | 94B | 32.61 | 33.12 | 34.45 | 56.35 | 29.55 | 32.65 | 28.35 | 35.30 |
| | 96B | 32.75 | 33.35 | 34.60 | 57.10 | 30.05 | 33.55 | 29.05 | 35.78 |
| | 98B | 32.78 | 33.28 | 34.58 | 57.45 | 30.15 | 32.95 | 28.55 | 35.68 |
| | 100B | 32.80 | 33.24 | 34.64 | 56.61 | 29.83 | 33.36 | 28.64 | 35.59 |
| | Mean ($\mu \pm \sigma$) | **32.74 ± 0.08** | **33.25 ± 0.10** | **34.57 ± 0.08** | **56.88 ± 0.49** | **29.90 ± 0.27** | **33.13 ± 0.40** | **28.65 ± 0.29** | **35.59 ± 0.21** |
| **+ 3-class Div.** | 94B | 32.35 | 32.88 | 34.95 | 58.40 | 33.20 | 32.95 | 28.40 | 36.16 |
| | 96B | 32.48 | 33.02 | 35.25 | 59.25 | 33.65 | 33.15 | 28.65 | 36.49 |
| | 98B | 32.50 | 32.96 | 35.15 | 58.65 | 33.15 | 33.65 | 28.95 | 36.43 |
| | 100B | 32.53 | 32.98 | 35.16 | 59.08 | 33.56 | 32.94 | 28.11 | 36.34 |
| | Mean ($\mu \pm \sigma$) | **32.47 ± 0.08** | **32.96 ± 0.06** | **35.13 ± 0.13** | **58.85 ± 0.39** | **33.39 ± 0.25** | **33.17 ± 0.33** | **28.53 ± 0.36** | **36.36 ± 0.14** |
| **+ 49-class Div.** | 94B | 33.18 | 33.25 | 35.30 | 58.32 | 33.12 | 34.22 | 28.55 | 36.57 |
| | 96B | 33.34 | 33.22 | 35.15 | 59.23 | 33.49 | 34.45 | 28.83 | 36.81 |
| | 98B | 33.58 | 32.95 | 35.45 | 58.79 | 33.55 | 34.58 | 29.07 | 36.84 |
| | 100B | 33.45 | 33.21 | 35.17 | 57.85 | 33.56 | 34.54 | 28.76 | 36.65 |
| | Mean ($\mu \pm \sigma$) | **33.37 ± 0.18** | **33.16 ± 0.14** | **35.27 ± 0.14** | **58.55 ± 0.60** | **33.43 ± 0.19** | **34.45 ± 0.14** | **28.80 ± 0.21** | **36.69 ± 0.13** |

ther enhances the architectural goal of separating general capabilities (shared) from specific ones (routed), potentially leading to cleaner modularization.

**2. Auxiliary-Loss-Free Balancing.** Some recent works propose replacing the auxiliary load-balancing loss ($\mathcal{L}_{LB}$) with a bias-based update mechanism (Wang et al., 2024b; DeepSeek-AI, 2024). We clarify that this mechanism targets *Load Balancing* (enforcing uniformity), whereas our $\mathcal{L}_{ED}$ targets *Expert Specialization* (enforcing divergence). These objectives are both important and non-conflicting. $\mathcal{L}_{ED}$ can be seamlessly integrated as an auxiliary loss alongside bias-based balancing to guide semantic specialization while the bias terms independently manage computational load.

**3. Unsupervised Router Regularization** Methods like ERNIE 4.5 (Baidu-ERNIE-Team, 2025) introduce regularizers to enforce orthogonality on the router weight matrices. In contrast, our method enforces divergence on the dynamic routing probability distributions. These approaches operate at different levels: ERNIE enhances the independence of the router's feature representation space in an unsupervised way, while $\mathcal{L}_{ED}$ utilizes data-driven supervision to ensure that semantic differences translate into structural differences in expert usage. Combining them would likely yield additive benefits, improving both representation quality and semantic specialization.

## K   COMPARISON WITH ERNIE 4.5 ROUTING ORTHOGONAL LOSS

We provide empirical evidence to confirm the compatibility of our approach with the MoE routing orthogonal loss introduced in the contemporaneous work, ERNIE 4.5 Baidu-ERNIE-Team (2025), as discussed in Appendix J. As illustrated in the training loss curves in Figure 9 and the perplexity on validation sets in Table 7, combining both objectives on the 8B-A0.8B model consistently exhibits additional gains compared to using the ERNIE 4.5 loss alone. These additive gains demonstrate that the two methods are not only compatible but mutually beneficial.

Table 7: Comparison of Validation Loss

| Models | zh_val_1000 | en_val_1000 | math_val_1000 |
|---|---|---|---|
| **Only ERNIE4.5 Loss** | 2.7974 | 3.1905 | 1.7556 |
| **Combine with Ours** | **2.7793** | **3.1905** | **1.7533** |

## L   GRANULARITY ABLATION: DOCUMENT-BASED VS. DOMAIN-BASED DIVERGENCE

To investigate the impact of granularity and validate our design, we conduct an ablation study by treating every document as a distinct class. This contrasts with our proposed Expert Divergence

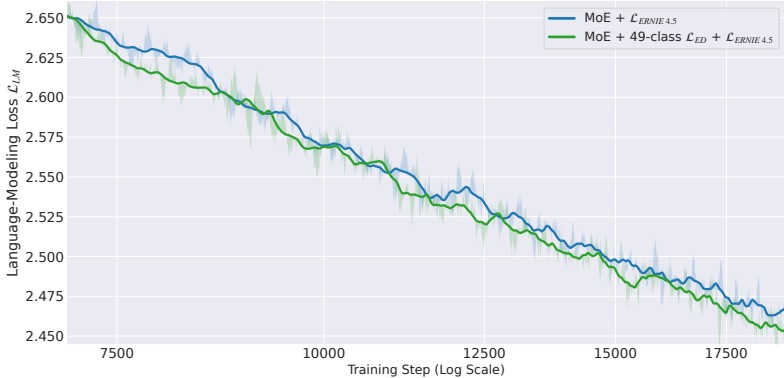

Figure 9: Loss Comparison of Using ERNIE 4.5 Loss and Adding the Expert Divergence Loss. Adding our loss shows additional loss gains.

method, which relies on source-based (3-class) and topic-based (49-class) domain label schemes. We trained an 8B-A0.8B model from scratch for 100B tokens under this configuration.

We report the mean and standard deviation across the final three checkpoints (96B, 98B, and 100B) in the table below. The ablation configuration yielded suboptimal results (34.88), even underperforming the Baseline MoE (35.12). This performance drop confirms the critical importance of **semantically meaningful domain labels** we used. Without semantic grounding, the divergence loss indiscriminately separates similar sequences in the expert distribution space, disrupting the intra-domain consistency required for stable expert specialization. This outcome reinforces our arguments in two key aspects:

1. It validates the necessity of domain classifiers. Since both coarse data sources (3-class) and granular document IDs (this ablation) underperform the 49-class setting, the investment in a domain classifier is proven essential.

2. It serves as a negative control to verify the validity of our performance gains. This ablation isolates the impact of the labeling strategy by keeping all other settings constant. The stark contrast—where our proposed method consistently improves performance while the document-level ablation consistently degrades it—confirms that our gains are significant and stem directly from the effective application of Expert Divergence Learning, rather than from random benchmark fluctuations.

Table 8: Ablation Study of Domain Labels

| Models | CEVAL | MMLU | CMMLU | ARC-e | ARC-c | RACE-m | RACE-h | Avg. |
|---|---|---|---|---|---|---|---|---|
| **Baseline MoE** | $32.78 \pm 0.22$ | $\mathbf{32.18} \pm 0.11$ | $\mathbf{34.11} \pm 0.34$ | $54.31 \pm 0.41$ | $\mathbf{31.78} \pm 0.32$ | $32.31 \pm 0.39$ | $\mathbf{28.36} \pm 0.37$ | $\mathbf{35.12} \pm 0.17$ |
| **+ One-class-per-doc Div.** | $32.67 \pm 0.33$ | $31.94 \pm 0.09$ | $33.92 \pm 0.39$ | $54.79 \pm 0.52$ | $30.51 \pm 0.22$ | $\mathbf{32.43} \pm 0.35$ | $27.92 \pm 0.21$ | $34.88 \pm 0.15$ |

## M    EXTENDED TRAINING TO 150B TOKENS

We scaled the training of the 15B-A1.5B models to 150B tokens (a 50% increase) to verify performance stability. By averaging statistics from the final three checkpoints, we observed two key trends: First, while extended training improved convergence for all models, our method consistently retained its lead. Second, a strict performance rank of "49-class ¿ 3-class ¿ Baseline" was evident on every dataset. These results validate that Expert Divergence Learning—particularly with fine-grained 49-class labels—delivers robust and comprehensive performance gains.

Table 9: Performance Comparison with Different Class Divisions

| Model | CEVAL | MMLU | CMMLU | ARC-e | ARC-c | RACE-m | RACE-h | Avg. |
|---|---|---|---|---|---|---|---|---|
| Baseline MoE | $34.54 \pm 0.25$ | $33.65 \pm 0.31$ | $35.89 \pm 0.49$ | $59.32 \pm 0.56$ | $34.35 \pm 0.54$ | $33.50 \pm 0.30$ | $28.34 \pm 0.53$ | $37.08 \pm 0.35$ |
| +3-class Div. | $35.43 \pm 1.24$ | $34.60 \pm 0.25$ | $36.52 \pm 0.33$ | $60.73 \pm 0.54$ | $35.14 \pm 0.41$ | $33.89 \pm 1.08$ | $28.69 \pm 0.37$ | $37.86 \pm 0.27$ |
| +49-class Div. | $36.11 \pm 0.55$ | $34.68 \pm 0.15$ | $36.58 \pm 0.32$ | $60.85 \pm 0.36$ | $35.25 \pm 0.22$ | $34.66 \pm 0.32$ | $28.73 \pm 0.32$ | $38.12 \pm 0.21$ |

