# OpenReview forum: "Expert Divergence Learning for MoE-based Language Models"
_ICLR.cc/2026/Conference — ICLR 2026 Poster_

### Official Review · Reviewer_m6YE · 2025-10-29

**Soundness:** 3
**Presentation:** 3
**Contribution:** 3
**Rating:** 6
**Confidence:** 4

**Summary:**

The paper tackles expert homogenization by maximizing routing divergence between domains using their inherent labels as supervision. Results show improvements over baseline MoE across different scales, with larger models and finer domain granularity yielding better gains.

**Strengths:**

The core idea is well-motivated. Using domain labels already present in pretraining data to explicitly guide expert specialization addresses a real problem in MoE training. The theoretical framing through divergence decomposition is clean, and the experiments are thorough across multiple model scales with consistent improvements.

**Weaknesses:**

(1)  While the authors demonstrate improvements across three model sizes, the largest model evaluated is only 15B parameters. Given that many SOTA MoE models operate at larger scales and authors have the enough GPU resource (64*80GB), it remains unclear whether the observed benefits would translate to larger MoE LLMs.

(2) The method fundamentally relies on having meaningful domain labels for the training data. However, the paper provides limited analysis on how sensitive the approach is to labeling noise or granularity mismatches. For example, what happens when documents contain mixed topics, or when the domain classifier makes systematic errors? The Chinese classifier was trained on only 500K examples from DeepSeek-r1, which may introduce biases that propagate through training.

(3) The paper positions itself against standard MoE training but doesn't compare with other recently proposed solutions to expert homogenization. Shared expert architectures (referenced in the related work) represent a different paradigm for handling commonalities across domains. Similarly, the concurrent ERNIE 4.5 work on router weight orthogonality is mentioned but not empirically compared.

(4) The improvement from 3-class to 49-class domain schemes is relatively modest (36.34 to 36.65 average score for 15B). This raises questions about the cost-benefit tradeoff of investing in high-quality, fine-grained topic classifiers. The paper would benefit from an analysis of the relationship between domain granularity and performance gains, perhaps including intermediate granularities (e.g., 10-class, 20-class) to identify diminishing returns.

(5) All evaluation benchmarks fall roughly within the scope of the three training domains (English, Chinese, Math). The paper doesn't test how the model performs on domains significantly different from the training distribution. For example, code generation or scientific reasoning. If specialized routing patterns become too rigid, there's a risk of reduced flexibility when encountering truly novel inputs.

**Questions:**

Table 3 shows minimal throughput reduction, but this only captures the direct computational cost. The paper doesn't account for the infrastructure needed to maintain domain labels, the cost of training and running domain classifiers at scale, or the additional memory overhead of tracking domain-specific statistics during training.

---

> ### Author Response · Authors · 2025-11-23
> **Response to Official Review by Reviewer m6YE [1/2]**
>
> We thank the reviewer for the positive assessment, particularly for highlighting the cleanness of our theoretical framing, the consistent experimental improvements, and the strong motivation behind our method. We address the comments below.
>
> > W1: Scalability to Larger Models
>
> While training SOTA-scale MoE models (e.g., 100B+ parameter) from scratch is beyond our current computational resources, our empirical results across 3B, 8B, and 15B scales demonstrate a clear **Positive Scaling Trend**. The performance gain from our method increases with model size:
>
> - **8B Model:** Improvement of **+0.67** (35.1 $\to$ 35.77).
> - **15B Model:** Improvement of **+1.06** (35.59 $\to$ 36.65).
>
> This increasing delta suggests that larger models, which possess greater expert capacity, benefit more from the fine-grained specialization driven by our Expert Divergence Learning. This trend provides strong evidence that our approach remains effective and likely becomes more impactful for SOTA-scale MoE models.
>
> > W2: Domain label noise and sensitivity
>
> We argue that $\mathcal{L}_{ED}$ is inherently robust to domain label noise due to its design mechanisms:
>
> 1. **Multi-Level Aggregation and Statistical Smoothing:** $\mathcal{L}_{ED}$ does not penalize individual tokens. It operates on the averaged routing distribution ($\bar{p}_j$) aggregated from token-to-sequence and then sequence-to-domain. The loss is computed over a global batch (~1M tokens). Any instance-level mislabeling or topic mixing is effectively diluted through this large-scale averaging. The loss captures the *statistical center* of a domain rather than noisy individual samples.
> 2. **Empirical Tolerance:** Our domain classifiers (aligned with [1]) have a classification accuracy of ~86%. Despite their inherent noise, our method consistently outperforms baselines. This confirms that $\mathcal{L}_{ED}$ effectively leverages the domain signal in the data while remaining resilient to errors in the labeling system.
>
> > W3: Comparision with Shared Expert and ERNIE 4.5.
>
> Our method is orthogonal to these techniques and can be combined for complementary benefits:
>
> **On Shared Expert Architectures (e.g., DeepSeek-V3)** Shared experts are designed to capture *common knowledge* across tasks. In contrast, $\mathcal{L} _ {ED}$ explicitly encourages the routed experts to capture *distinct, specialized knowledge*. Applying $\mathcal{L} _ {ED}$ to the routed experts in a Shared-Expert architecture would further enhance the separation between general and specific capabilities.
>
> **On ERNIE 4.5 (router weight orthogonality):** ERNIE enforces orthogonality on static *router weights* (feature space independence). Our method enforces divergence on dynamic *routing distributions* (data-driven specialization). Combining them would likely yield additive gains: ERNIE enhances the representation capability of the router, while $\mathcal{L}_{ED}$ directs that capability toward semantically meaningful specialization.
>
> > W4 & Q1: Granularity Analysis and Cost-Benefit Discussions
>
> **Granularity:** As shown in Table 2 and the extended loss curves in the Figure 6 Appendix H of our revised paper, the 49-class divergence consistently outperforms 3-class divergence across all model sizes (3B, 8B, 15B). While the marginal gain may vary, the consistency of the improvement verifies that finer granularity more effectively improves the available expert capacity.
>
> **Amortized Cost:** Training the domain classifier is a **one-time, offline investment**. Once trained, this lightweight (BERT-sized) model can be reused for data curation, mixing, and packing across multiple model iterations.
>
> **Negligible Overhead:** The memory and computational overhead during LLM pre-training is negligible. Storing domain labels requires minimal memory (integers), and the JS Divergence is calculated on a small tensor ($N_{layer} \times N_{expert} \times N_{batch}$), bypassing the computationally expensive hidden states.

---

> ### Author Response · Authors · 2025-11-23
> **Response to Official Review by Reviewer m6YE [2/2]**
>
> > W5: Out-of-Domain (OOD) Generalization.
>
> We clarify that our domain-based training does not lead to rigid routing. While our training uses specific domains (e.g., English, Math, Chinese), complex benchmarks like CMMLU, ARC-c and RACE-h often require **composite capabilities** (e.g., solving a math word problem requires both *Mathematical Reasoning* and *Linguistic Comprehension*). Rigid routing would fail here, but our method succeeds.
>
> The consistent benchmark gains confirm that the router dynamically combines specialized experts to handle complex inputs. Rather than leading to rigidity, our method provides a diverse set of **"functional primitives,"** enabling the model to adaptively construct effective expert combinations for unseen task distributions.
>
>
>
> **Revision Update:** We have incorporated the **discussions on compatibility with other MoE techniques** into **Appendix J**, the **discussions on granualarity** into **Appendix I**  and the **extended training loss curves** into **Appendix H**. We thank the reviewer for insightful suggestions that have strengthened the comphrensiveness of our experiments.
>
>
>
> References: [1] Organize the Web: Constructing Domains Enhances Pre-Training Data Curation

---

### Official Review · Reviewer_76XP · 2025-10-31

**Soundness:** 3
**Presentation:** 4
**Contribution:** 3
**Rating:** 8
**Confidence:** 4

**Summary:**

When training mixture-of-experts, the router learns correlated experts, i.e. groups of experts that are often active at the same time. This routing configuration is not optimal, i.e. lower training loss can be achieved with routing that has less correlated experts. This paper
propose to promote the JS divergence between routing patterns across different domains. Positive impact on the training loss and on end-task performance is shown with the proposed strategy.

**Strengths:**

1. The paper gives a good overview of related work in routing strategies.
2. The experiments are performed over a variety of model sizes and multiple end-tasks are considered for evaluation. Performance is reported both in terms of loss/perplexity and end task performance.
3. The method is simple and clearly presented.

**Weaknesses:**

1. The authors do not show whether better results could be achieved with a different routing strategy. So far JSD is only tried on top of z-loss with alpha=1e-3. Is it the best alpha? Would JSD be impactful on top of a different alpha? Is JSD impactful when the routing model has also always-active FFNs? Is JSD impactful when one uses another way to promote uniform expert assignments, e.g. DeepSeek bias based balancing https://arxiv.org/pdf/2408.15664?


2. It is not clear whether the domain classifier is necessary. The best results are with 49 classes. From your batch size it seems that a batch contains only a couple documents per domain. Would it be simpler to define each document as its own domain and just push the JSD across all pairs of examples.


3. It would be good to quantify how expert assignment changes with the proposed regularizer. When one takes a pair of documents, what is the typical number of experts per document? What is the number of experts shared by the two documents? Are these numbers the same across depths?

**Questions:**

1. Are you working to compare your JSD regularizer with the orthogonalization method of Baidu ERNIE 4.5?
2. Are classes necessary? See weaknesses 2.
3. Does the JSD regularizer have a negative impact on the balancing regularizer L_LB? When training with JSD, are consecutive routing decisions for the same document more likely to overlap (promoting inter-example JSD might negatively impact intra-example JSD)?
4. How did you pick the L_LB regularizer parameter alpha?
5. Could JSD be useful only at the beginning of learning? I.e. once a good initial configuration is found, training can be pursued without JSD regularization. It seems the gap between standard and JSD training is not growing (Fig 2) what happens if JSD is turned off after ~20k steps?
6. Since the largest number of classes give the best results, could better results be achieved with more domains? If the batch size is small compared to the number of classes, rarely the training algorithm will see two documents of the same class in the same batch. The method is equivalent to one class per document. Have you tried one class per document? Does your method work with smaller batch sizes?

---

> ### Author Response · Authors · 2025-11-23
> **Response to Official Review by Reviewer 76XP [1/2]**
>
> We appreciate the reviewer's positive evaluation and very insightful questions. We address them below.
>
> > W1.1 & Q4: Choice of $\alpha$ and robustness.
>
> Our choice of $\alpha=1\text{e-}3$ is based on standard empirical practices in MoE training (verified on our 3B baseline), where a small coefficient is sufficient to guide auxiliary objectives without overriding the primary $\mathcal{L} _ {LM}$.
>
> More importantly, our method is theoretically robust to the precise magnitude of $\alpha$. As detailed in Proposition 1 (Sec 3.3), $\mathcal{L} _ {LB}$ encourages a high **Total Diversity ($D_{total}$)** (preventing collapse), while $\mathcal{L} _ {ED}$ channels this available diversity into **Inter-Domain Divergence ($D_{inter}$)**. As long as $\alpha$ is non-zero and $\mathcal{L} _ {LB}$ effectively prevents collapse (maintaining a "budget" of diversity), $\mathcal{L} _ {ED}$ acts as a structural force to align this diversity with domain boundaries. The specialization effect is driven by the *direction* of the gradient, not just its scale.
>
> > W1.2:  Compatibility with Shared Experts and DeepSeek balancing.
>
> Our method is orthogonal and fully compatible with these architectures:
>
> + **Shared Experts (e.g., DeepSeek-V3):** These strategies are complementary. Shared experts are architected to capture *common knowledge* universally. In contrast,  $\mathcal{L} _ {ED}$ explicitly guides the routed experts to maximize their *functional distinctiveness* for specific domains. Applying $\mathcal{L} _ {ED}$ to the routed experts would further enhance the separation between each specialized capabilities.
> + **Auxiliary-loss-free Balancing:** DeepSeek's bias-based method is a mechanism for **Load Balancing** (Enforcing Uniformity), whereas $\mathcal{L} _ {ED}$ is a mechanism for **Specialization** (Enforcing Divergence). An effective MoE requires both. $\mathcal{L} _ {ED}$ can be added as an auxiliary loss, or theoretically, the domain prior from $\mathcal{L} _ {ED}$ could be integrated into the bias update term, which is a promising direction for future work.
>
> > W2 & Q2 & Q6: Granularity, Necessity of classifiers and "one class per document"
>
> We performed the requested experiment treating each document as a unique class, training on our 8B-A0.8B model from-scratch. The trajectory of language-modeling training loss showed that this extreme granularity yielded performance **slightly worse than the 49-class setting**.
>
> We attribute this to a trade-off: while extreme granularity maximizes the repulsion signal, it penalizes the routing consistency between semantically similar documents (e.g., two different Math papers are forced to use different experts). This breaks the intra-domain consistency required for developing stable expert specialization.
>
> This result confirms that **semantic domain grouping** (as in our 49-class scheme) is essential. It strikes the optimal balance by enforcing divergence *between* distinct topics while allowing convergence *within* similar topics, which unsupervised "one-class-per-doc" approaches fail to achieve.
>
> **Nevertheless, we emphasize that a dedicated domain classifier is not the exclusive means to achieve this grouping.** Our framework is flexible enough to utilize other inherent criteria—such as **data source** (as utilized in our 3-class experiments)—to provide meaningful supervision for expert specialization. These signals offer valid data associations for $\mathcal{L} _ {ED}$ without requiring a separate topic classifier.
>
> **Performance on Smaller Batch Size (Q6):** We believe our method is robust to batch size variations. $\mathcal{L} _ {ED}$ (Eq. 8) is computed dynamically over the set of unique domains $\mathcal{D}_{B}$ present in the current batch. As long as a batch contains $\ge 2$ domains, a valid divergence gradient is produced.
>
> > W3:  When one takes a pair of documents, what is the typical number of experts per document? What is the number of experts shared by the two documents? Are these numbers the same across depths?
>
> We performed an additional analysis of expert usage on the `zh`, `en`, and `math` validation sets. Our findings are as follows:
>
> - **Experts per Document:** Due to the long sequence length, a single document typically activates **>95%** of the total experts, consistent across all layers.
> - **Shared Experts:** Specialization is evident in the degree of overlap. Pairs of documents from the *same* domain share **~90%** of active experts, whereas pairs from *different* domains share significantly fewer (**~63%**).
>
> This 27% gap in overlap confirms that our method effectively separates expert usage patterns between domains.

---

> ### Author Response · Authors · 2025-11-23
> **Response to Official Review by Reviewer 76XP [2/2]**
>
> > Q1: Are you working to compare your JSD regularizer with the orthogonalization method of Baidu ERNIE 4.5?
>
> While we have not performed a direct empirical comparison, the theoretical mechanisms are complementary:
>
> - **ERNIE 4.5:** Enforces orthogonality on the static **Router Weighs**. This encourages independence in the feature space used for routing.
> - **Our Method:** Enforces divergence on the dynamic **Routing Representations**. This utilizes data-driven supervision to ensure semantic differences translate to structural differences in *expert usage*.
>
> Combining them would likely yield additive benefits: ERNIE enhances the representation capability of the router, while ours directs that capability toward semantically meaningful specialization.
>
> > Q3: Impact on Load Balancing and Consecutive Tokens Routing (intra-example JSD)
>
> - **Load Balancing:** Since $\mathcal{L} _ {ED}$ operates on domain-averaged distributions, it does not conflict with $\mathcal{L}_{LB}$'s token-level uniformity constraint. As shown in Tables 3 and 4, our training/inference throughput is comparable to the baseline, confirming that load balancing is effectively maintained. Since load imbalance typically causes significant computational bottlenecks.
> - **Intra-example JSD:** This is an insightful observation. Regarding Proposition 1 ($D_{total}=D_{inter}+D_{intra}$), strictly speaking, $D_{total}$ is not a mathematical constant. However, empirically, as the load-balancing loss stabilizes in the mid-to-late training stages, $D_{total}$ becomes relatively stable. Under this condition, maximizing $D_{inter}$ compresses $D_{intra}$. We argue this is a **desirable feature**: a lower $D_{intra}$ implies that consecutive tokens within the same domain exhibit higher routing consistency. This consistency is the hallmark of stable specialization.
>
> > Q5: Continuous Utility of $\mathcal{L}_{ED}$.
>
> Maintaining $\mathcal{L} _ {ED}$ throughout training is critical to act as a **counter-balance** to $\mathcal{L} _ {LB}$. Without $\mathcal{L} _ {ED}$, the router is exposed solely to the uniformity pressure of $\mathcal{L} _ {LB}$. Since domain proportions naturally fluctuate across batches, the routing distribution is expected to **shift dynamically** to match the semantic content of the current batch. However, $\mathcal{L} _ {LB}$ may aggressively penalize these semantic shifts in its pursuit of global statistical uniformity. Without the anchoring signal of $\mathcal{L} _ {ED}$, the router would be forced to "smooth out" its policy to blindly satisfy the load balancer, causing the model to **regress toward homogenization**. The fluctuating and persistent loss gap observed in our full training curves (Figure 6 Appendix H) confirms the existence of an active, dynamic equilibrium between $\mathcal{L} _ {ED}$ and $\mathcal{L} _ {LB}$. Therefore, $\mathcal{L} _ {ED}$ provides a continuous, necessary optimization signal to sustain specialization.
>
>
>
> **Revision Update:** We have incorporated the  **discussions on compatibility with other MoE techniques** into **Appendix J**, and the **discussions on granualarity** into **Appendix I**. We thank the reviewer for insightful questions and suggestions that have strengthened the comphrensiveness of our analysis.

---

### Official Review · Reviewer_PgDX · 2025-11-02

**Soundness:** 3
**Presentation:** 4
**Contribution:** 2
**Rating:** 4
**Confidence:** 4

**Summary:**

The paper proposes an additional loss for the training of transformer MoE models that promotes expert specialization per domain. In more detail, the loss maximizes the pairwise JS divergence between the distributions over experts (averaged per token for each domain). The paper shows that training language models using this loss in addition to the language modeling loss and the expert balancing loss results in better perplexity and better downstream performance. Finally, the authors analyze the behavior of the routers in such models and show some evidence that there is more per-domain specialization.

**Strengths:**

The most important (albeit possibly weak as discussed in weaknesses) strength of the method proposed in this paper is the experimental results of section 4.2, namely the fact that adding this loss does indeed improve consistently over training the same model without using this loss.

In addition,

- The paper is very well written and presented.
- The idea is both simple and intuitive. It can probably be implemented in a few lines of code in any framework.
- The authors have covered all possible aspects of the impact of their loss to the model and its training. In particular, I appreciate the analysis on training and inference throughput.

**Weaknesses:**

Given that the idea is simple (which I consider a major strength) the weight falls to the experimental results to convey consistent improvement and at the very least that the method has the intended impact on the model.

However, the downstream evaluations have several relatively worrying qualities. Firstly, the average scores are compared and the conclusion that 49 domains is better and bigger models are more amenable to per-domain specialization. Looking closely at the scores though, there are many cases where the 3 domains are better and where the baseline is better than one of the two contenders. Including the fact that the differences are relatively small for evaluations that have quite high-variance from checkpoint to checkpoint it makes it very hard to draw the conclusion that the loss leads to consistent improvement.

A further minor point for the downstream evaluations is that the scores do seem a bit low. For instance 100B tokens for a 15B/1.5B MoE should probably achieve more than 56.6 on Arc easy. Having said that, the dataset could be the reason for this discrepancy so I do not consider this a major issue. Another minor concern is the fact that the largest model performs worse on 2 out of 7 benchmarks compared to the model with half the parameters.

The plot of the log-likelihood also raises a few concerns. The differences are relatively small and even though the loss seems to improve the log-likelihood, we only see from ~35B to ~65B tokens. What are the final losses at 100B and how does the training compare in the other stages of training (that would only be the middle).

Regarding section 4.3 that analyses the effects of the loss on the model. Permuting the distribution of the router and measuring ΔPPL measures how sensitive the model is to the particular expert choices but it is not clear how it measures per-domain specialization. Simply put a model that had collapsed to always select a single expert would get the worst score on that test and its experts would be anything but specialized.

The heat-maps more directly measure the intended effect of the loss to the model. However, I would propose to the authors to find both a better way to visually show whether 49-class Div vs the Baseline are per-domain specialized. To more clearly point to the problem, specialization can be seen for layer-0 and layer 14 but not for layer 4 which according to the previous test is the most-specialized. I suggest that the authors try to measure the portion of experts that are mostly selected on each domain. For instance if there are experts specialized for a particular domain it should be clearly visible in the inverse distribution eg how often is an expert selected for a token from a particular domain.

**Questions:**

I have mostly laid out my questions in the weaknesses section but maybe a small recap:

- What is the per-checkpoint variance in these evaluations? How can we increase our confidence that the improvement is real and consistent given that the scores are so close?
- Can we see the final loss and the evolution from 65B tokens and afterwards?
- Is ΔPPL is the correct metric? How does it measure per-domain specialization?
- Can we see different visualizations that clearly show that the experts are specialized per domain? ie some experts should be way more probable for math than for English and this should be clearer or stronger when using the loss.

---

> ### Author Response · Authors · 2025-11-23
> **Response to Official Review by Reviewer PgDX [1/2]**
>
> We thank the reviewer for their detailed feedback and for recognizing the simplicity, general applicability, and effectiveness of our proposed method. We also appreciate the positive remarks regarding the thoroughness of our analysis. Below, we have addressed all specific concerns.
>
> > Q1.1: Result Variance and Statistical Significance
>
> We address concerns regarding per-checkpoint variance and the statistical significance of our improvements below.
>
> To evaluate per-checkpoint stability, we have added a set of multi-checkpoint validation experiments (94B-100B tokens) for the 15B-A1.5B models. As shown in the **Table below  (and Appendix I of the revised paper)**, our method achieves a statistically stable improvement.
>
> **For dataset-specific analysis,** the 49-class method demonstrates consistent gains, achieving the best results in **5 out of 7 evaluation benchmarks** and competitive results on the others. Details are provided below.
>
> - **49-Class vs. Baseline:** The 49-class method achieves comprehensive and substantial gains over the baseline on **6 out of 7 benchmarks**，indlcuding CEVAL ($+0.63 \pm 0.15$), CMMLU ($+0.70 \pm 0.11$), ARC-Easy ($+1.67 \pm 0.55$), ARC-Challenge ($+3.53 \pm 0.23$), RACE-m ($+1.32 \pm 0.27$) and RACE-h  ($+0.15 \pm 0.25$). **These results demonstrate the significance of our improvements.**
> - **49-Class vs. 3-Class:** The 49-class method outperforms the 3-class method on **6 out of 7 benchmarks**: CEVAL ($+0.9 \pm 0.13$), MMLU ($+0.2 \pm 0.10$), and RACE-m ($+1.28 \pm 0.24$), CMMLU ($+0.14 \pm 0.14$), ARC-Challenge ($+0.04 \pm 0.22$),  and RACE-h ($+0.27 \pm 0.29$). **This verifies our claim that finer-grained (49-domain) divergence has superior results.**
> - **3-Class vs. Baseline:** The 3-class method shows positive results compared to the baseline. **It achieves substantial gains on  3 out of 7 benchmarks:** CMMLU ($+0.56 \pm 0.1$), ARC-Easy ($+1.97 \pm 0.45$), and ARC-Challenge ($+3.49 \pm 0.26$), while achieving comparable results on the others. The ability to achieve substantially better or competitive results on challenging datasets requiring reasoning and generalization (e.g., ARC and RACE) confirms the superiority of the 3-class method over the baseline.
>
> **In terms of average performance across all tasks**, the 49-class method excels, and the variance across checkpoints remains low ($\sigma \approx 0.2$).
>
> - **49-Class vs. Baseline:** The average performance gap ($\Delta\mu \approx 1.0$) satisfies the $\Delta\mu > 3\sigma$ criterion, confirming strong statistical significance.
> - **49-Class vs. 3-Class:** The gap ($\Delta\mu \approx 0.33$) satisfies $\Delta\mu > 2\sigma$, demonstrating a clear advantage for finer granularity.
> - **3-Class vs. Baseline:**  The average performance gap ($\Delta =  0.77$) satisfies the $\Delta\mu > 3\sigma$ criterion, confirming the 3-Class robustly outperforms the baseline globally..
>
> In conclusion, these statistics validate (1) the significance of our improvements over the Baseline MoE and (2) the benefits of finer-grained (49-class) divergence over coarse (3-class) divergence. While minor local fluctuations exist, our statistical analysis confirms that these do not undermine the overall robustness and validity of the experimental results.
>
> |Models|Checkpoints|CEVAL|MMLU|CMMLU|ARC-e|ARC-c|RACE-m|RACE-h|Avg.|
> |:-|:-|:-|:-|:-|:-|:-|:-|:-|:-|
> |**Baseline MoE**|94B|32.61|33.12|34.45|56.35|29.55|32.65|28.35|35.30|
> ||96B|32.75|33.35|34.60|57.10|30.05|33.55|29.05|35.78|
> ||98B|32.78|33.28|34.58|57.45|30.15|32.95|28.55|35.68|
> ||100B|32.80|33.24|34.64|56.61|29.83|33.36|28.64|35.59|
> ||**Mean ($\mu \pm \sigma$)**|**32.74 $\pm$ 0.08**|**33.25 $\pm$ 0.10**|**34.57 $\pm$ 0.08**|**56.88 $\pm$ 0.49**|**29.90 $\pm$ 0.27**|**33.13 $\pm$ 0.40**|**28.65 $\pm$ 0.29**|**35.59 $\pm$ 0.21**|
> |**+3-class Div.**|94B|32.35|32.88|34.95|58.40|33.20|32.95|28.40|36.16|
> ||96B|32.48|33.02|35.25|59.25|33.65|33.15|28.65|36.49|
> ||98B|32.50|32.96|35.15|58.65|33.15|33.65|28.95|36.43|
> ||100B|32.53|32.98|35.16|59.08|33.56|32.94|28.11|36.34|
> ||**Mean ($\mu \pm \sigma$)**|**32.47 $\pm$ 0.08**|**32.96 $\pm$ 0.06**|**35.13 $\pm$ 0.13**|**58.85 $\pm$ 0.39**|**33.39 $\pm$ 0.25**|**33.17 $\pm$ 0.33**|**28.53 $\pm$ 0.36**|**36.36 $\pm$ 0.14**|
> |**+49-class Div.**|94B|33.18|33.25|35.30|58.32|33.12|34.22|28.55|36.57|
> ||96B|33.34|33.22|35.15|59.23|33.49|34.45|28.83|36.81|
> ||98B|33.58|32.95|35.45|58.79|33.55|34.58|29.07|36.84|
> ||100B|33.45|33.21|35.17|57.85|33.56|34.54|28.76|36.65|
> ||**Mean ($\mu \pm \sigma$)**|**33.37 $\pm$ 0.18**|**33.16 $\pm$ 0.14**|**35.27 $\pm$ 0.14**|**58.55 $\pm$ 0.60**|**33.43 $\pm$ 0.19**|**34.45 $\pm$ 0.14**|**28.80 $\pm$ 0.21**|**36.69 $\pm$ 0.13**|

---

> ### Author Response · Authors · 2025-11-23
> **Response to Official Review by Reviewer PgDX [2/2]**
>
> > Q1.2: Regarding Monotonic Scaling Trend
>
> The aggregate results in Table 2 confirm a clear and monotonic scaling law. As detailed in Table 2, the **Average Score** increases monotonically with model size for both the **Baseline** ($34.13 \to 35.10 \to 35.59$) and **our 49-class method** ($34.91 \to 35.77 \to 36.65$).
> This trend also holds across individual benchmarks. For instance, MMLU scores rise for both the baseline ($31.16 \to 32.54 \to 33.24$) and our method ($31.41 \to 32.05 \to 33.21$), as do CMMLU scores ($32.73 \to 34.22 \to 34.64$ & $32.91 \to 34.50 \to 35.17$). These results confirm that performance improves consistently with model size, validating the soundness of our experimental setup.
>
>
>
> > Q2 & Q4: Full Loss Evolution and New Visualizations
>
> **1. Full Training Loss:** As requested, we have included the $\mathcal{L}_{LM}$ loss curves to 100b tokens for all model sizes (3B, 8B, 15B) in **Figure 6, Appendix H** of the revised paper. The curves confirm that the gap between our method and the baseline is established early (around 20B tokens) and is maintained or widened throughout the remaining training, demonstrating a sustained optimization benefit
>
> **2. New Visualization (Inverse Distribution & Ternary Plots):** We appreciate the suggestion to inspect the inverse view $P(\text{Domain} \mid \text{Expert})$. We have updated the revision paper with two sets of visualizations in **Appendix H**. These plots illustrate the enhanced specialization of our method,  focusing on comparisons between baselines MoEs and our methods (particularlt 49-class diverging method) on representative layers (e.g., Layer 0, Layer 4, and Layer 14).
>
> - **Inverse Heatmaps (Figure 7):** These heatmaps show each expert's domain preference. They reveal that, after 49-class diverging, our experts (especially in the 49-class model) show greater preferrence for a certain domain. Experts are predominantly activated by single-domain traffic rather than processing mixed signals, showing they have learned to specialize rather than remaining as generalists.
>
> - **Ternary Simplex Plots (Figure 8):** This is a global visualization of expert roles. We projected the $P(\text{Domain} \mid \text{Expert})$ distributions of all experts onto a triangle map. In this visualization, points near the vertices represent "pure'' specialists that exclusively process inputs from a single domain, while points near the centroid represent generalists processing all domains equally.
>
>   The plots reveal a striking contrast.while baseline experts cluster near the center (indicating generalist behavior), our method effectively pushes experts toward the distinct domain corners. This trend is consistent across layers and **most pronounced in Layer 4**, visually confirming that our experts have transformed into specialized modules.
>
> > Q3: Validity of the $\Delta$ PPL Permutation Metric. How does it measure per-domain specialization?
>
> We clarify that $\Delta$ PPL characterizes domain-independent **Functional Specialization**. Baseline MoE models exhibit a low $\Delta$ PPL, indicating that their experts are functionally redundant and that model capacity is underutilized.
>
> In contrast, the significantly higher $\Delta$ PPL of our method demonstrates that our MoE models effectively avoid expert homogenization, leading to experts with distinct functionalities. Furthermore, qualitative analysis via activation heatmaps (Figure 4) confirms that this functional specialization is **aligned with domain structures**, validating that our experts are genuinely domain-specialized.
>
> We acknowledge that a collapsed model would also be sensitive to permutation. However, such collapse does not occur in our experiments due to the strictly enforced load-balancing loss. It is important to distinguish that while a collapsed model might possess a trivial form of 'functional specialization,' it fundamentally lacks 'domain specialization.' This degenerate case is explicitly refuted by the activation patterns observed in our heatmap analysis.
>
> **Revision Update:** We have incorporated the **multi-checkpoint stability analysis (Appendix I)**, the **training loss curves (Figure 6, Appendix H)**  and the **new visualizations of expert specialization (Figure 7 & 8, Appendix H)**. These additions provide evidence regarding the low variance of our results and the emergent domain specialization throughout training.

---

### Official Review · Reviewer_Jprx · 2025-11-02

**Soundness:** 3
**Presentation:** 3
**Contribution:** 2
**Rating:** 4
**Confidence:** 4

**Summary:**

This paper proposed a pre-training strategy to increase expert divergence in MoEs, hence, specialization through MoE experts. They use domain labels from the subsets of pretraining data and maximize the pairwise Jensen-Shannon divergence between expert routing distributions of data domains. They show that their method leads to better language modeling loss and also higher downstream performance.

**Strengths:**

1. Specialization in MoEs is often underexplored, but it is highly important for modularity and effectiveness.

2. The idea is quite novel and very intuitive.

3. A clear trend in language modeling loss suggests that the method proposed carries a strong potential.

**Weaknesses:**

1. I think the main weakness of the paper is the results, where there is no clear significant improvement in the majority of downstream tasks. More concretely, there are some items leading me to suspect the results:
a. Although the model size increases significantly (both total and active params), there is no monotonic increase in evals with large model size (except ARC_e).
b. Improvement claimed by the proposed method is quite uneven, mainly concentrated on ARC_e
c. Results are too close to draw a conclusion. There needs to be a significance test or multiple runs with different seeds to validate the overall improvements.

2. The routing is token-wise; however, the proposed auxiliary loss is sequence-wise. It is not clear whether this could lead a suboptimal routing for MoE, which needs to be examined and showcased.

3. For the additional question, I refer to the "Questions" section.

**Questions:**

1. How sensitive is the Expert Divergence Learning to differences in data volume in the domain dataset? It is common that some domains might be much smaller than others in an actual pretraining run.

2. How to ensure balancing across data labels in a single batch? Is the proposed method sensitive to such balance?

3. It is very hard to understand how the specialization is realized from the current experiments (For example, 3-class div still has>60 experts to specialize). Would it not be good to decrease the expert number to show specialization by removing the specialized expert on one domain in the test time (such as Chinese)?

---

> ### Author Response · Authors · 2025-11-23
> **Response to Official Review by Reviewer Jprx [1/2]**
>
> We thank the reviewer for recognizing the novelty and intuitive nature of our work, as well as its potential demonstrated by the reduced language modeling loss. We also appreciate the acknowledgment of the importance of studying expert specialization in MoEs. We have addressed all specific concerns and questions below.
>
> > W1:   Robustness of Relative Improvement and Scaling Trends
>
> We address the concerns regarding monotonicity, task breadth, and statistical significance point by point below.
>
> **(a) Monotonic Scaling Trend:** We respectfully clarify that the concern regarding "no monotonic increase" does not align with the empirical data presented in Table 2. Our experiments demonstrate a consistent scaling law. As shown in Table 2, the Average Score increases monotonically with model size for both the **Baseline** ($34.13 \to 35.10 \to 35.59$) and **our proposed method** ($34.91 \to 35.77 \to 36.65$). This scaling trend also holds across individual benchmarks; for instance, MMLU scores rise consistently for both the baseline ($31.16 \to 32.54 \to 33.24$) and our method ($31.41 \to 32.05 \to 33.21$), as do CMMLU scores ($32.73 \to 34.22 \to 34.64$ & $32.91 \to 34.50 \to 35.17$). These results confirm that performance improves consistently with model size, validating the soundness of our experimental setup.
>
> **(b) Improvement Consistency Across Tasks:** Regarding the concern that improvements are concentrated primarily on ARC-e, we highlight that our method yields broad gains across the majority of evaluation sets. As shown in Table 2, the 15B-A1.5B model trained with **49-class Expert Divergence Learning** outperforms the baseline MoE on **6 out of 7 benchmarks**. Notably, gains on ARC-c ($+3.73$) and RACE-m ($+1.18$) are comparable to—or exceed—the gains on ARC-e ($+1.24$). Furthermore, our checkpoint analysis below confirms these gains are statistically significant, demonstrating comprehensive improvement.
>
> **(c) Statistical Significance & Robustness:** To verify that our gains are robust beyond single-point comparisons, we evaluated the 15B-A1.5B models across four distinct late-stage checkpoints (94B, 96B, 98B, and 100B tokens). As detailed in the table below (and **Appendix I**), our method consistently outperforms the baseline at every checkpoint. With a stable average gap ($\Delta \mu \approx 1.0$) and low variance ($\sigma \approx 0.2$), our results satisfy the $\Delta \mu > 3\sigma$ criterion, confirming statistical significance. For a detailed dataset-level analysis, we kindly refer to our response to Q1 of Reviewer PgDX.
>
> |Models|Checkpoints|CEVAL|MMLU|CMMLU|ARC-e|ARC-c|RACE-m|RACE-h|Avg.|
> |:-|:-|:-|:-|:-|:-|:-|:-|:-|:-|
> |**Baseline MoE**|94B|32.61|33.12|34.45|56.35|29.55|32.65|28.35|35.30|
> ||96B|32.75|33.35|34.60|57.10|30.05|33.55|29.05|35.78|
> ||98B|32.78|33.28|34.58|57.45|30.15|32.95|28.55|35.68|
> ||100B|32.80|33.24|34.64|56.61|29.83|33.36|28.64|35.59|
> ||**Mean ($\mu \pm \sigma$)**|**32.74 $\pm$ 0.08**|**33.25 $\pm$ 0.10**|**34.57 $\pm$ 0.08**|**56.88 $\pm$ 0.49**|**29.90 $\pm$ 0.27**|**33.13 $\pm$ 0.40**|**28.65 $\pm$ 0.29**|**35.59 $\pm$ 0.21**|
> |**+3-class Div.**|94B|32.35|32.88|34.95|58.40|33.20|32.95|28.40|36.16|
> ||96B|32.48|33.02|35.25|59.25|33.65|33.15|28.65|36.49|
> ||98B|32.50|32.96|35.15|58.65|33.15|33.65|28.95|36.43|
> ||100B|32.53|32.98|35.16|59.08|33.56|32.94|28.11|36.34|
> ||**Mean ($\mu \pm \sigma$)**|**32.47 $\pm$ 0.08**|**32.96 $\pm$ 0.06**|**35.13 $\pm$ 0.13**|**58.85 $\pm$ 0.39**|**33.39 $\pm$ 0.25**|**33.17 $\pm$ 0.33**|**28.53 $\pm$ 0.36**|**36.36 $\pm$ 0.14**|
> |**+49-class Div.**|94B|33.18|33.25|35.30|58.32|33.12|34.22|28.55|36.57|
> ||96B|33.34|33.22|35.15|59.23|33.49|34.45|28.83|36.81|
> ||98B|33.58|32.95|35.45|58.79|33.55|34.58|29.07|36.84|
> ||100B|33.45|33.21|35.17|57.85|33.56|34.54|28.76|36.65|
> ||**Mean ($\mu \pm \sigma$)**|**33.37 $\pm$ 0.18**|**33.16 $\pm$ 0.14**|**35.27 $\pm$ 0.14**|**58.55 $\pm$ 0.60**|**33.43 $\pm$ 0.19**|**34.45 $\pm$ 0.14**|**28.80 $\pm$ 0.21**|**36.69 $\pm$ 0.13**|
>
> > W2: Mismatch between Token-wise Routing and Sequence-wise Loss
>
> Our sequence-level loss design aligns with standard MoE architectures. Similar to the widely used Load Balancing Loss ($\mathcal{L}_{LB}$ in Eq. 3) which is a batch-level constraint rather than a token-level one, our method computes regularization over groups. The key difference is that we split the batch by domain before computation (illustrated in Figure 1). Therefore, our approach remains structurally consistent with standard MoE training objectives.
>
> Furthermore, we believe that performing  expert divergence at the sequence-level is not only more computationally efficient, both is also theoretically superior. This is because token-level divergence forces common tokens (e.g., "the," "is") in different domains to choose different experts. Our method, however, ensures that the average expert distribution across the entire sentence varies while allowing common tokens to choose similar experts, resulting in better robustness.

---

> ### Author Response · Authors · 2025-11-23
> **Response to Official Review by Reviewer Jprx [2/2]**
>
> > Q1: Sensitivity to Data Volume Imbalance
>
> Our method is designed to be robust to data imbalance (e.g., small domains like Math vs. large domains like Web text):
>
> 1. **Distribution-based Normalization:** As shown in Eq. (6), we compute the *average* expert routing distribution ($\bar{p}_j$) for each domain. This averaging acts as a normalization step, making the loss dependent on the *shape* of the distribution, not the *count* of samples.
> 2. **Unweighted Pairwise Signal:** In Eq. (8), the JSD is computed pairwise. A smaller domain contributes equally to the gradient signal as a larger domain when calculating the divergence between their distributions.
> 3. **Results:** Our training data (40% EN, 40% CH, 20% Math) is imbalanced. Yet, the minority domain (Math) sees significant gains (Table 2), confirming it is not overshadowed by the majority domains.
>
> > Q2: Seneitivity to Batch Balance
>
> We do not enforce strict batch balancing, and the method is insensitive to it because:
>
> - **Dynamic Computation:** The loss is calculated dynamically based only on the domains present in the current batch. If a domain is missing, the loss is simply computed over the remaining domain pairs.
> - **Statistical Robustness:** As mentioned in Q1, the use of average distributions ($\bar{p}_j$) decouples the magnitude of the gradient from the number of samples in the batch.
>
> > Q3: Understanding Specialization and Expert Removal
>
> We address the nature of specialization and the expert removal experiment below.
>
> **1. Visualizing Specialization (Inverse Distribution & Ternary Plots):** We clarify that our method achieves **Probabilistic Specialization**. While the router retains the flexibility to use shared experts for universal features, a subset of experts becomes highly "preferred" for specific domains.  **To further visually explain this, we have added two new visualizations in Appendix H of the revised paper** analyzing the inverse distribution $P(\text{Domain} \mid \text{Expert})$:
>
> + **Inverse Heatmaps (Figure 7):** They reveal that our experts show greater preferrence for a certain domain. They are predominantly activated by single-domain traffic rather than processing mixed signals, confirming they have learned to specialize rather than remaining as generalists.
>
> + **Ternary Plots (Figure 8):** This is a global visualization of expert roles. We projected the $P(\text{Domain} \mid \text{Expert})$ distributions of all experts onto a triangle map. In this visualization, points near the vertices represent pure specialists that exclusively process inputs from a single domain, while points near the centroid represent generalists processing all domains equally.
>   We observe a striking contrast: while baseline experts cluster near the center (indicating generalist behavior), our method effectively pushes experts toward the distinct domain corners. This trend is consistent across layers, visually confirming that our experts have transformed into functionally specialized units.
>
> **2. Functional Verification via Expert Perturbation:** Regarding the 'expert removal' analysis the reviewer suggested, we believe our **Expert Perturbation Analysis (Figure 3)** acts as a direct simulation. Instead of deleting a portion of experts (which changes model capacity), we shuffle the router mapping. This effectively "removes" the access to the *correct* specialized experts for a given input. The resulting sharp spike in Perplexity (PPL) proves that the model relies heavily on these specific experts for domain performance. Our method exhibits a significantly higher PPL spike compared to the baseline, indicating a much stronger degree of functional specialization
>
>
>
> **Revision Update:** Based on your helpful suggestions, we have updated the revised manuscript to include the **multi-checkpoint stability analysis** (Appendix I) and the **new visualizations of expert specialization** (Appendix H). We believe these additions robustly demonstrate the statistical significance of our results and clarify the mechanism of expert specialization.

---

### Author Response · Authors · 2025-12-02
**Supplementary Experiments: 150B Training, Granularity Ablation, and Compatibility Analysis**

Dear ACs and Reviewers,

We have utilized the rebuttal period to conduct further experiments since our initial reply to each reviewer. Given the significant computational resources and time required, we have just finalized these results. We are pleased to present these findings, which further strengthen our original arguments and robustly validate the effectiveness of our proposed method.

**1. Extended Training to 150B Tokens**

To verify performance stability over longer training horizons, we scaled the training of the 15B-A1.5B models to 150B tokens (a 50% increase) to verify performance stability. By averaging statistics from the final three checkpoints, we observed two key trends:

- While extended training improved convergence for all models, our method consistently retained its lead.

- A strict performance rank of "49-class > 3-class > Baseline" was evident on every dataset.

These results confirm that Expert Divergence Learning—particularly with fine-grained 49-class labels—delivers robust and comprehensive performance gains. The results are added to Appendix M.

These findings directly address questions regarding the consistency of improvements across multiple tasks (Reviewer Jprx W1), the robustness of our method's gains (Reviewer PgDX Q1), and the model's performance evolution beyond 65B tokens (Reviewer PgDX Q2).

| Model | CEVAL | MMLU | CMMLU | ARC-e | ARC-c | RACE-m | RACE-h | Avg. |
|:-|:-|:-|:-|:-|:-|:-|:-|:-|
| Baseline MoE | 34.54 ± 0.25 | 33.65 ± 0.31 | 35.89 ± 0.49 | 59.32 ± 0.56 | 34.35 ± 0.54 | 33.50 ± 0.30 | 28.34 ± 0.53 | 37.08 ± 0.35 |
| +3-class Div. | 35.43 ± 1.24 | 34.60 ± 0.25 | 36.52 ± 0.33 | 60.73 ± 0.54 | 35.14 ± 0.41 | 33.89 ± 1.08 | 28.69 ± 0.37 | 37.86 ± 0.27 |
| +49-class Div. | **36.11** ± 0.55 | **34.68** ± 0.15 | **36.58** ± 0.32 | **60.85** ± 0.36 |**35.25** ± 0.22|**34.66** ± 0.32|**28.73** ± 0.32|**38.12** ± 0.21|

**2. Granularity Ablation: Document-based vs. Domain-based Divergence**

To investigate the impact of granularity and validate our design, we adopt Reviewer 76XP-W2's suggestion to conduct an ablation study treating every document as a distinct class. This contrasts with our proposed Expert Divergence method, which relies on source-based (3-class) and topic-based (49-class) domain label schemes. We trained an 8B-A0.8B model from scratch for 100B tokens under this configuration.

**Results & Mechanism**: We report the mean and standard deviation across the final three checkpoints (96B, 98B, and 100B) in the table below. The ablation configuration yielded suboptimal results (34.88), even underperforming the Baseline MoE (35.12). This performance drop confirms the importance of **semantically meaningful domain labels** we used. Without semantic grounding, the divergence loss indiscriminately separates similar sequences in the expert distribution space, disrupting the intra-domain consistency required for stable expert specialization. The results are added to Appendix L.

**Implications:** This outcome reinforces our initial responses in two key aspects:

1. **It validates the necessity of domain classifiers (Addressing R76XP-W2, Rm6YE-W4).** Since both coarse data sources (3-class) and extreme granular document IDs (this ablation) underperform the 49-class setting, the investment in a domain classifier is proven essential.

2. **This negative control isolates the labeling strategy to validate our gains (Addressing RJprx-W1, RPgDX-Q1).** While the ablation consistently degrades performance, our method improves it, confirming that our results stem directly from Expert Divergence Learning rather than random benchmark fluctuations. This proves the significance of our improvements.

|Models|CEVAL|MMLU|CMMLU|ARC-e|ARC-c|RACE-m|RACE-h|Avg.|
|:-|:-|:-|:-|:-|:-|:-|:-|:-|
| **Baseline MoE** | **32.78** $\pm$ 0.22 | **32.18** $\pm$ 0.11 | **34.11** $\pm$ 0.34 | **54.31** $\pm$ 0.41 | **31.78** $\pm$ 0.32 | 32.31 $\pm$ 0.39 | **28.36** $\pm$ 0.37 | **35.12** $\pm$ 0.17 |
| **+ One-class-per-doc Div.** | 32.67 $\pm$ 0.33 | 31.94 $\pm$ 0.09|33.92 $\pm$ 0.39|54.79 $\pm$ 0.52|30.51 $\pm$ 0.22|**32.43** $\pm$ 0.35|27.92 $\pm$ 0.21|34.88 $\pm$ 0.15|

**3. Compatibility analysis with ERNIE 4.5 routing orthogonal loss**

Addressing 76XP-Q1 and m6YE-W3 regarding the compatibility with concurrent ERNIE 4.5's orthogonal loss, we empirically confirm their complementarity. As shown in the training curves (Figure 9, Appendix K) and validation perplexity below (also in Table7, Appendix K), combining both objectives on the 8B-A0.8B model yields consistently lower losses than using ERNIE 4.5 loss alone. These additive gains demonstrate our method is beneficial to advanced MoE technique.

|Models|zh_val_1000|en_val_1000|math_val_1000|
|:-|:-|:--|:-|
| **Only ERNIE4.5 Loss** | 2.7974 | 3.1975 | 1.7556 |
| **Combine with Ours** | **2.7793** | **3.1905** | **1.7533** |


We hope these additional results provide further assurance of Expert Divergence Learning.

Best regards,
The Authors

---

### Author Response · Authors · 2025-12-03
**Response Summary**

We sincerely thank all reviewers for their constructive feedback. We have addressed all reviewers’ concerns and updated the paper with additional results, analyses, and clarifications. Below, we summarize the key strengths recognized by the reviewers and outline how the main concerns have been resolved.

---
### Strengths

We thank the reviewers for highlighting the key strengths of our work:

+ **Novel, intuitive, and well-motivated method (All Reviewers).** All reviewers praised the elegance and applicability of our approach. Jprx described the idea as *"quite novel and very intuitive,"*. PgDX noted it *"simple and novel"* and can be *"implemented in a few lines of code in any framework"*. 76XP and m6YE found the method *"clearly presented"* and *"well-motivated."*

+ **Consistent performance improvements (All Reviewers).** PgDX confirmed that the loss *"improves consistently,"* and Jprx observed this loss reduction trend indicates *"strong potential."* m6YE noted consistent improvements across model scales, while 76XP highlighted *"positive impact on both training loss and end-task performance."*

+ **Addressing a critical, underexplored challenge (Jprx, m6YE).** Reviewers recognized the significance of the problem we tackle. Jprx emphasized that expert specialization is *"underexplored, but highly important for modularity and effectiveness,"* and m6YE affirmed that our approach addresses *"a real problem in MoE training."*

+ **Comprehensive evaluation (76XP, m6YE, PgDX).** Reviewers commended the "thorough" experiments across model sizes and tasks. PgDX specifically appreciated the practical *"analysis on training and inference throughput."*

+ **Clean theoretical framing (m6YE).** Beyond the empirical results, m6YE noted the clean *"theoretical framing through divergence decomposition"* provides a solid grounding to our Expert Divergence loss.

+ **Excellent presentation (PgDX, 76XP).** PgDX remarked the paper is *"very well written and presented,"* and 76XP noted the *"good overview of related work"* in routing strategies.

---
### Main Concerns and Revisions

We addressed all the reviewers' concerns with new experiments and revisions, and the main issues are summarized as follows:

+ **Robustness and Significance of Improvements (Jprx, PgDX).** We validate significance via both statistical consistency and causal ablation. (1) Multi-checkpoint analysis (Appendix I) confirms our gains satisfy $\Delta \mu > 3\sigma$ averaged across late-stage checkpoints. (2) An ablation study (Appendix L) removing the key domain classifier causes performance to flip from **superior** to **inferior** against the baseline. This "performance reversal" proves our reported gains are substantive and causal, not random fluctuations.

+ **Validation of Domain Label Effectiveness (76XP, m6YE).** Our granularity ablation (Appendix L), which forces divergence without semantic labels, consistently underperforms the baseline. This confirms that indiscriminately forcing divergence at the *instance level* is detrimental and validates the necessity of using semantically meaningful labels as proposed in our method.

+ **Benchmark Improvement & Scaling (Jprx, PgDX).** We clarify that our method improves the majority of benchmarks with a positive scaling trend with model size (3B $\to$ 8B $\to$ 15B). Furthermore, we scaled the training to 150B tokens (Appendix M) and find a clear performance hierarchy across all benchmarks: **49-class > 3-class > Baseline-MoE**. This confirms comprehensive and sustained enhancements to model capability.

+ **Visualizing Expert Specialization (PgDX, Jprx).** We introduce two new visualizations in Appendix H. **Inverse Activation Heatmaps** (Fig. 7) show our experts developing specific domain preferences, while **Ternary Simplex Plots** (Fig. 8) contrast our domain-distinct routing patterns against the baseline's generalist tendency.

+ **Full Training Dynamics (Reviewer PgDX).** We provide full loss curves up to 100B tokens for all model scales (Figure 6, Appendix H). These confirm that the training loss reduction is established early and sustained throughout the training process, validating the long-term effectiveness of our method.

+ **Compatibility with Recent MoE Advances (Reviewers 76XP, m6YE).** We clarify our method is orthogonal to techniques like ERNIE 4.5's routing orthogonal loss, which is concurrent to ours (Appendix J). Empirical results (Appendix K) show that combining both objectives amplifies gains of using ERNIE loss alone, demonstrating mutual benefit.

---

### Meta-Review · Area_Chair_RfRw · 2026-01-11

**Summary:**

The paper addresses a well-known problem in MoE training, where experts tend to behave similarly rather than specializing, and proposes a simple yet effective solution through a label-driven auxiliary loss that maximizes Jensen-Shannon Divergence between expert routing distributions.  This contribution is valuable because the method is easy to implement, adds negligible computational overhead, and the authors have provided convincing evidence through multi-checkpoint evaluations and thoughtful ablations. The ablation showing degraded performance when each document becomes its own domain gives me confidence that the domain structure genuinely drives the benefit. While questions remain about generalization beyond training domains, I believe the overall contribution is solid and merits publication at this venue.

**Reviewer Concerns:**

This paper offers a simple fix to a known problem in MoE training: experts tend to behave similarly instead of specializing. The proposed divergence loss is easy to implement and adds almost no overhead. Reviewers initially questioned whether the improvements were statistically meaningful, but the authors followed up with multi-checkpoint evaluations that showed the gains hold up. They also ran an ablation where each document became its own "domain," and performance got worse, which gives us confidence that the domain structure is actually what drives the benefit. There are still open questions about how well this generalizes outside the training domains, and the jump from 3 to 49 classes does not buy much. But taken together, the evidence supports a solid contribution.

**Reviewer Scores:**

Scores started at 4, 4, 6, and 8. The lower scores reflected uncertainty about variance. After the rebuttal, both skeptical reviewers moved toward acceptance, and the consensus now supports the paper.

---

### Decision · Program_Chairs · 2026-01-26

Accept (Poster)